

# Reconciling the differences between OMI-based and EPA AQS in situ NO₂ trends

Ruixiong Zhang[1], Yuhang Wang[1], Charles Smeltzer[1], Hang Qu[1], William Koshak[2], K. Folkert Boersma[3,4]

[1]School of Earth and Atmospheric Sciences, Georgia Institute of Technology, Atlanta, Georgia, USA
[2]NASA-Marshall Space Flight Center, National Space Science & Technology Center, 320 Sparkman Drive, Huntsville, Alabama, USA
[3]Meteorology and Air Quality Group, Wageningen University, the Netherlands
[4]Royal Netherlands Meteorological Institute, De Bilt, the Netherlands

*Correspondence to*: Yuhang Wang (yuhang.wang@eas.gatech.edu)

**Abstract.** With the improved spatial resolution than earlier instruments and more than ten years of service, tropospheric NO₂ retrievals from the Ozone Monitoring Instrument (OMI) have led to many influential studies on the relationships between socioeconomic activities and NOx emissions. This study focuses on how to improve OMI NO₂ retrievals for more reliable trend analysis. We retrieve OMI tropospheric NO₂ vertical column densities (VCDs) and obtain the NO₂ seasonal trends over the United States, which are compared with coincident in situ surface NO₂ measurements from the Air Quality System (AQS) network. The Mann-Kendall method is applied to derive the NO₂ seasonal and annual trends for four regions at coincident sites during 2005-2014. The OMI-based NO₂ seasonal relative trends are generally biased high compared to the in situ trends by up to 3.7% yr⁻¹, except for the underestimation in the Midwest and Northeast during Dec-Jan-Feb (DJF). We improve the OMI retrievals for trend analysis by removing the ocean trend, using the MODerate-resolution Imaging Spectroradiometer (MODIS) albedo data in air mass factor (AMF) calculation, and applying a lightning flash filter to exclude lightning affected OMI NO₂ retrievals. These improvements result in close agreement (within 0.3% yr⁻¹) between in situ and OMI-based NO₂ regional annual relative trends. Thus, we recommend future studies to apply these procedures to ensure the quality of satellite-based NO₂ trend analysis, especially in regions without reliable long-term in situ NO₂ measurements. We derive optimized OMI-based NO₂ regional annual relative trends using all available data for the West (-2.0%±0.3 yr⁻¹), the Midwest (-1.8%±0.4 yr⁻¹), the Northeast (-3.1%±0.5 yr⁻¹), and the South (-0.9%±0.3 yr⁻¹). The OMI-based annual mean trend over the contiguous United States is -1.5%±0.2 yr⁻¹. It is a factor of 2 lower than that of the AQS in situ data (-3.9%±0.4 yr⁻¹); the difference is mainly due to the fact that the locations of AQS sites are concentrated in urban and suburban regions.

## 1 Introduction

Nitrogen dioxide (NO₂) is an air pollutant. At high concentrations, it aggravates respiratory diseases and can lead to acid rain formation (e.g., Lamsal et al., 2015). It is also a key player to produce another pollutant, ozone (O₃), through photochemical



reactions in the presence of Volatile Organic Compounds (VOCs) under sunlight. Tropospheric $NO_2$ is emitted both anthropogenically and naturally (e.g., Gu et al., 2016). Anthropogenic fossil fuel combustions and biomass burnings emit mostly nitrogen monoxide (NO) under high temperature, which is later oxidized by $O_3$ into $NO_2$. Major natural $NO_2$ sources include lightning and soils.

Surface $NO_2$ concentrations are regulated by the U.S. Environmental Protection Agency (EPA) through the National Ambient Air Quality Standards (NAAQS). $NO_2$ is measured routinely at the EPA Air Quality System (AQS) sites (Demerjian, 2000). Although the AQS network continually provides valuable hourly $NO_2$ measurements, AQS sites are mostly located in urban and suburban regions, leaving large regions of rural areas unmonitored. Satellite data provide a better spatial coverage than the in situ measurements.

Several satellites were launched to monitor tropospheric $NO_2$ vertical column densities (VCDs), such as the SCanning Imaging Absorption spectroMeter for Atmospheric CHartographY (SCIAMACHY), the Global Ozone Monitoring Experiment–2 (GOME-2), and the Ozone Monitoring Instrument (OMI). For trend analysis, the tropospheric $NO_2$ products from OMI surpass the others for a relatively high spatial resolution and over one decade of continuous operation (Boersma et al., 2004; Boersma et al., 2011). Thus, OMI $NO_2$ retrievals are widely applied in $NO_2$ and NOx emission trend studies (e.g.,
Lin et al., 2010, 2011; Castellanos et al., 2012; Russell et al., 2012; Gu et al., 2013; Lamsal et al., 2015; Lu et al., 2015; Tong et al., 2015; Cui et al., 2016; Duncan et al., 2016; de Foy et al., 2016a, 2016b; Krotkov et al., 2016; Liu et al., 2017). Tong et al. (2015) reported that the reduction rates calculated from OMI $NO_2$ VCDs and AQS surface $NO_2$ data at eight cities were -35% and -38% from 2005 to 2012, respectively. Lamsal et al. (2015) also found the divergence between the annual trends inferred from the two datasets, i.e. -4.8% $yr^{-1}$ vs -3.7% $yr^{-1}$ during 2005-2008, and -1.2% $yr^{-1}$ vs -2.1% $yr^{-1}$ during 2010-2013.

To understand how the retrieval procedure affects the resulting OMI derived trends and their differences from those derived from the surface AQS measurements, we utilize a regional 3-D chemistry transport model (CTM), a radiative transfer model (RTM), and the Mann-Kendall method (Mann, 1945; Kendall, 1948) to calculate OMI-based $NO_2$ seasonal relative trends during Dec-Jan-Feb (DJF), Mar-Apr-May (MAM), Jun-Jul-Aug (JJA), and Sept-Oct-Nov (SON) (Section 2). To reconcile with the AQS based regional $NO_2$ trends, we find that three procedures are essential to ensure the quality of trend analysis
using OMI tropospheric $NO_2$ VCDs, including the ocean trend removal, the MODerate-resolution Imaging Spectroradiometer (MODIS) albedo update in calculating the air mass factors (AMFs), and the lightning filter (Section 3.1). With these procedures implemented, the differences between OMI-based and AQS in situ annual relative trends are within 0.3% $yr^{-1}$ of coincident measurements for all the four regions. Finally, we estimate the OMI-based annual relative trends across the nation in Section 3.2. Conclusions are given in Section 4.



## 2 Methods

### 2.1 EPA AQS surface NO$_2$ measurements

The in situ surface NO$_2$ measurements from the U.S. EPA AQS network are used in this research. Sites with a continuous measurement gap of more than 50 days are removed and the observations of 140 remaining cites are used (Fig. 1). The AQS chemiluminescent analyzers are equipped with molybdenum converters to measure ambient NO$_2$ concentrations. These analyzers are known to have high biases, since the converters are not NO$_2$ specific and they measure some fractions of peroxyacetyl nitrate, nitric acid and organic nitrates (Demerjian, 2000; Lamsal et al., 2008). In addition to chemiluminescent analyzers, several NO$_2$ specific photolytic instruments were deployed since 2013. By utilizing the data from both chemiluminescent and photolytic measurements at coincident sites during the overpassing time of OMI, we calculate the observed NO$_2$ concentration ratio between both measurements in Fig. S1 in the Supplement. The ratio peaks at 2.3 in June and decreases to 1.3 in November, indicating that the chemiluminescent analyzers overestimate by 27%-132% than photolytic instruments. This finding is in agreement with Lamsal et al. (2008). We correct the chemiluminescent NO$_2$ data by the observed ratio assuming that the inter-annual change is small. This correction may contribute to the differences between in situ and OMI based absolute NO$_2$ trends but do not significantly affect the relative trends (since the correction is canceled out in computing relative trends). In this study, we only examine the relative trends and therefore the analysis results are not affected by the uncertainties in the in situ NO$_2$ measurement corrections.

### 2.2 REAM model

We use a 3-D Regional chEmical trAnsport Model (REAM) in the simulation of NO$_2$ profiles. REAM has widely been used in atmospheric NO$_2$ studies, including vertical transport (Choi et al., 2005; Zhao et al., 2009a; Zhang et al., 2016a), emission inversions (Zhao et al., 2009b; Yang et al., 2011; Gu et al., 2013, 2014, 2016), and regional and seasonal variations (Choi et al., 2008a, 2008b). The model has a horizontal resolution of 36 km with 30 vertical layers in the troposphere, 5 vertical layers in the stratosphere, and a model top of 10 hpa. In this study, the domain of REAM is about 400 km larger on each side than the contiguous United States (CONUS). Meteorology inputs driving transport process are simulated by the Weather Research and Forecasting model (WRF) assimilations constrained by National Centers for Environmental Prediction Climate Forecast System Reanalysis (NCEP CFSR, Saha et al., 2010) 6-hourly products. The KF-eta scheme is used for sub-grid convective transport in WRF (Kain and Fritsch,1993). We run the WRF model with the same resolution as in REAM but with a domain 10 grids larger on each side than that of REAM. REAM updates most of the meteorology inputs every 30 minutes while those related to convective transport and lightning parameterization are updated every 5 minutes. The chemistry mechanism expands that of a global CTM GEOS-Chem (V9-02) with aromatics chemistry (Bey et al., 2001; Liu et al., 2010, 2012a, 2012b; Zhang et al., 2017). For consistency, the GEOS-Chem (V9-02) simulation with 2° × 2.5° resolution is used to generate initial and boundary conditions for chemical tracers.





Anthropogenic emissions of NOx and other chemical species are from the U.S. National Emission Inventory 2008 prepared using the Sparse Matrix Operator Kernel Emission (SMOKE) model. Biogenic emissions are simulated online using the Model of Emissions of Gases and Aerosols from Nature (MEGAN) algorithm (v2.1, Guenther et al., 2012). We parameterize lightning emitted NOx as a function of convective mass flux and Convective Available Potential Energy (CAPE) (Choi et al., 2005).

NOx production per flash is set to 250 moles NO per flash, and the emissions are distributed vertically following the C-shaped profiles by Pickering et al. (1998). For recent model evaluations of REAM with observations, we refer readers to Zhang et al. (2016a, 2016b), Cheng et al. (2017), and Zhang et al. (2017).

## 2.3 OMI-based NO$_2$ VCDs

We retrieve the tropospheric NO$_2$ VCDs using the tropospheric slant column densities (SCDs) from the Royal Dutch

Meteorological Institute (KNMI) Dutch OMI NO$_2$ product (DOMINO v2, Boersma et al., 2011). OMI onboard the Aura satellite was launched in July 2004 and is still active. OMI overpasses the equator at about 13:30 Local Time (LT) and obtains global coverage with a 2600 km viewing swath spanning 60 rows. It has a ground level spatial resolution up to 13 km x 24 km (at nadir). SCDs are retrieved by matching a modeled spectrum to an observed top-of-atmosphere reflectance with the Differential Optical Absorption Spectroscopy (DOAS) technique within a fitting window of 405-465nm. The stratospheric

portion of SCDs are estimated and subsequently removed with a global CTM TM4 with stratospheric ozone assimilation (Dirksen et al., 2011). Deriving tropospheric VCDs from the remaining tropospheric SCDs requires the calculation of AMFs. Being an optically thin gas, tropospheric AMF for NO$_2$ can be calculated from AMF for each vertical layer ($AMF_l$) weighted by NO$_2$ VCDs at the corresponding layer ($x_l$) (Boersma et al., 2004), as shown in equation (1).

$$\text{tropospheric AMF} = \frac{tropospheric\ SCD}{tropospheric\ VCD} = \frac{\int AMF_l x_l dl}{\int x_l dl} \qquad (1)$$

As the vertical distribution of NO$_2$ is usually unknown, we typically substitute $x_l$ by an a priori profile ($x_{l,apriori}$) from a CTM. $AMF_l$ is the sensitivity of NO$_2$ SCD to VCD at a given altitude (Eskes and Boersma, 2003), and is computed using the Double Adding KNMI (DAK) RTM (Lorente et al., 2017). As a result, the retrieved tropospheric NO$_2$ VCD computation depends on the a priori NO$_2$ vertical profile, the surface reflectance, the surface pressure, the temperature profile, and the viewing geometry (Boersma et al., 2011). Previous studies have addressed the sources of uncertainties in NO$_2$ retrievals,

including surface reflectance resolutions (Russell et al., 2011; Lin et al., 2014), lightning NOx (Choi et al., 2005a; Martin et al., 2007; Bucsela et al., 2010), a priori CTM uncertainties (Russell et al., 2011; Heckel et al., 2011; Lin et al., 2012; Laughner et al., 2016), surface pressure and reflectance anisotropy in rugged terrain (Zhou et al., 2009), cloud and aerosol radiance (Lin et al., 2014, 2015), and boundary layer dynamics (Zhang et al., 2016a). The NO$_2$ VCD trend analysis is particularly sensitive to the first two factors and we will discuss these in the following sections.



AMFs are derived using the pre-computed altitude-dependent AMF lookup table, which is generated by the DAK RTM. We use the $NO_2$ profiles from REAM, temperature and pressure from CSFR, viewing geometry and cloud information from DOMINO v2 product. We use the REAM results of 2010 to avoid the uncertainty introduced by yearly variation of $NO_2$ profiles. The yearly variations of meteorology and anthropogenic emission changes have little impact on trend analysis results

using OMI data (Lamsal et al., 2015). We use the surface reflectance from DOMINO v2 product as default (Kleipool et al., 2008), and update it using a surface reflectance product with a higher temporal resolution (Section 2.3.2). The derived tropospheric $NO_2$ VCD relative trends are referred as "Standard".

### 2.3.1 Ocean trend removal

For trend and other analyses of OMI tropospheric VCDs, the data of anomalous pixels must be removed. The row anomaly

initially occurred in June 2007 and subsequently in later years affected rows 26-40 (Schenkeveld et al., 2017). Additional anomalies can be found in some years in rows 41-55. For trend analysis from 2005-2014, we exclude rows 26-55, consistent with our understanding of the row anomaly (Schenkeveld et al., 2017), and following the flagging in the DOMINO v2 data product. In addition, the data of coarse spatial resolution from rows 1-5 and rows 56-60 are also excluded, as suggested by Lamsal et al. (2015). Furthermore, we exclude OMI data with cloud fraction > 0.3 to minimize retrieval uncertainties due to

clouds and aerosols (Boersma et al., 2011; Lin et al., 2014).

Fig. 2a shows that there is an apparent increasing trend of the averaged tropospheric SCDs in the remote ocean region (Fig. 2b) with minimal marine traffic. This trend reflects the increase in the magnitude of the stripes (step-wise SCD variability from one row to another) in time, which originates from the use of a constant (2005-averaged) solar irradiance reference spectrum in the DOAS spectral fits throughout the mission and the weak increase of noise in the OMI radiance measurements (Boersma,

personal communication, 2017; Zara et al., 2018). Fig. 2a shows that there is a positive annual trend of $1.75 \pm 0.45 \times 10^{13}$ molecules $cm^{-2}$ $yr^{-1}$. The ocean trend is insensitive to the region selection in the remote North Pacific (varies within 10%). We only analyze OMI tropospheric column trends over the CONUS for grid cells with 2005-2014 averaged VCDs > $1 \times 10^{15}$ molecules $cm^{-2}$, which tends to minimize the effect of the background noise. However, removing this background ocean (absolute) trend has a non-negligible effect in reducing the OMI relative trend (Fig. 1). We refer to such derived (relative)

trend data as "Ocean". An alternative method is to subtract monthly SCDs of the remote ocean region from the OMI tropospheric SCD data. Although the end results are essentially the same as the trend removal method, noises are added to the SCD data (Fig. 2a), making it more difficult to understand the effects of the MODIS albedo update and the lightning filter (next sections). We therefore choose to use the (absolute) trend removal method here.

### 2.3.2 MODIS albedo update

The albedo data used to calculate the $AMF_l$ in "Standard" and "Ocean" versions of trend analysis are from the DOMINO v2 products, which are the climatology of averaged OMI measurements during 2005-2009 with a spatial resolution of $0.5° \times 0.5°$ (Kleipool et al., 2008) and is valid for 440 nm. We recalculate the $AMF_l$ using the MODIS 16-day MCD43B3 albedo product





with 1km spatial resolution, which combines data from both MODIS onboard Aqua and Terra satellites (Schaaf et al., 2002; Tang and Zhang, 2007). Aqua and Terra have an equatorial overpassing time of 13:30 LT and 10:30 LT, respectively. The band 3 (459nm-479nm) is used to match the $NO_2$ fitting window (405nm-465nm). The albedo is spatially integrated to the geometry of OMI pixels and is temporally interpolated to match OMI overpassing dates. In order to maintain the consistency

of the DOMINO retrieval algorithm (Boersma et al., 2011), we only use the MODIS data to improve the temporal variations of albedo data used in the retrieval. We scale the MODIS albedo data such that the mean albedo during 2005-2009 is the same as the OMI climatology at 0.5°×0.5°. We recalculate OMI tropospheric VCDs using the MODIS albedo data as described. We recalculate the relative OMI trend and remove the ocean (absolute) trend (Section 2.3.1). We refer to this version of OMI relative trend data as "MODIS".

**2.3.3 Lightning event filter**

Over North America, lightning is a major source of NOx in the free troposphere and its simulations in CTMs are uncertain (e.g., Zhao et al., 2009a; Luo et al., 2017). The large temporospatial variations of lightning NOx make it difficult to compute satellite based $NO_2$ trends by changing the vertical distributions of $NO_2$ affecting the AMF calculation (e.g., Choi et al., 2008b; Lamsal et al., 2010) and the SCD values. Given the difficulty to simulate lightning NOx accurately across different years, we

use a lightning filter to remove potential effects of lightning NOx on the basis of the flash rate observations of cloud-to-ground (CG) lightning flash data detected by the National Lightning Detection Network™ (NLDN) (Cummins and Murphy, 2009; Rudlosky and Fuelberg, 2010). NLDN only reports the ground point of a CG lightning flash, while the CG lightning flash can extend horizontally to tens of kilometers. A CG lightning flash can affect the $NO_2$ retrievals not only in the model grid cell where the CG lightning is located but also the nearby model grid cells. The atmospheric lifetime of NOx in the free troposphere

can be up to 1 week. Therefore, we exclude the OMI $NO_2$ data within a radius of 90km of the NLDN-reported CG lightning location (about two model grid cells around the grid cell where the CG lightning is located) for a period of 72 hours after the lightning occurrence. Since lightning usually occur along the track of a thunderstorm, the 90 km radius is more a constraint on lightning NOx effects across the track. The extended period of 72 hours is to ensure that we exclude data affected by lightning NOx. Figure 4 shows the distribution of the number of days of 2005-2014 with lightning detection. The Southwest monsoon

and the South regions have more lightning days than the other areas. While there are fewer lightning flashes in the Northeast than the South (Fig. 3), large amounts of lightning NOx can be produced by high flash ratios of severe thunderstorms and they can be transported northward from the South to the Northeast (Choi et al., 2005). We therefore further filter OMI $NO_2$ data in the Northeast on the basis of CG lightning flash rates in the South. If the average CG flash rate in the South exceeds the 95[th] percentile value of the NLDN observations, which is 0.035 flash $km^{-2}$ $day^{-1}$ (Fig. S2 in the Supplement), we exclude in the

analysis the Northeast OMI data in the following 72 hours. Excluding the OMI data based on CG lightning data implicitly removes the data affected by cloud-to-cloud lightning collocated with CG lightning. The lightning filter removes about 2%, 27%, 20%, and 19% of OMI data, which are coincident with AQS data, for the West, the Midwest, the Northeast and the South, respectively. We refer to this version of OMI relative trend data as "Lightning filter".





## 3 Results and discussion

We group the analysis results into different regions: (a) West, (b) Midwest, (c) Northeast, and (d) South (Fig. 1), following the regional divisions by the United States Census Bureau. To make a fair comparison between the in situ and OMI-based trends, we only use spatially and temporally coincident in situ and OMI $NO_2$ observations in Section 3.1. We apply the Mann-Kendall method to calculate the relative trend of $NO_2$ for each season, i.e. DJF, MAM, JJA, and SON, during 2005-2014. We construct the uncertainties of the trends with a confidence level of 95%. Note that when we compare in situ and OMI-based trends, the lightning filter also removes in situ $NO_2$ data, which are coincident with the OMI $NO_2$ data affected by lightning. This leads to slightly different in situ $NO_2$ trends between Fig. 4 and Fig. 6 (Section 3.2.3). We first compute the trends using the "Standard" OMI VCD data. The ocean trend removal, MODIS albedo update, and lightning filter are then added in sequence to compute three different OMI-based $NO_2$ trends to compare to the AQS in situ results. A subtlety in the comparison is that the coincident data change when the lightning filter is applied. As a result, the AQS in situ results in this set of comparison differ from those in the other three sets.

### 3.1 In situ and "Standard" OMI-based trends

Fig. 4 shows that both AQS in situ and "Standard" OMI-based seasonal relative trends are negative for all seasons across the regions. OMI-based trends generally underestimate the decreasing trends by up to 3.7% $yr^{-1}$ except for the large overestimation in the Midwest and the Northeast regions during DJF. The overestimates in these two regions are 3.0% $yr^{-1}$ and 1.1% $yr^{-1}$, respectively. On average, the differences between OMI-based and in situ seasonal relative trends are 1.6% $yr^{-1}$, -0.3% $yr^{-1}$, 1.0% $yr^{-1}$, and 1.4% $yr^{-1}$ for the West, the Midwest, the Northeast, and the South regions, respectively. Note that the relative trends are calculated using coincident measurements for the comparisons. The $NO_2$ relative trends from both datasets are expected to be close on a regional basis where surface emissions of NOx dominate the observed surface concentrations and tropospheric VCDs of $NO_2$. The focus of this work is to reconcile the difference between AQS in situ and OMI-based trends, which will be discussed in the following sections.

### 3.1.1 Improvement due to ocean trend correction

After removing the ocean trend as discussed in Section 2.3.1, the OMI-based $NO_2$ decreasing trends are more pronounced as shown in Fig. 4 ("Ocean", blue line) by 0.1-0.9% $yr^{-1}$. The regional relative trends have different sensitivities to the ocean trend removal due to different tropospheric VCDs levels. In general, the discrepancies between OMI-based and in situ trends are reduced except for the Midwest and the Northeast regions during DJF, which are already biased low. The averaged differences between OMI-based and in situ seasonal relative trends for the West, the Midwest, the Northeast, and the South regions are 1.2% $yr^{-1}$, -1.1% $yr^{-1}$, 0.4% $yr^{-1}$, and 1.0% $yr^{-1}$. Only in the Midwest region, removing the ocean trend enlarges the difference due to the large winter bias.



### 3.1.2 Improvement due to MODIS albedo update

The adoption of the up-to-date MODIS albedo (Section 2.3.2) greatly reduces the difference of relative trends in the Midwest during DJF from -3.6% yr$^{-1}$ ("Ocean") to 1.3% yr$^{-1}$ ("MODIS"), the improvement of DJF trend difference is more moderate from -1.7% to 0.5% (Fig. 4). There are no significant changes of the comparisons in other regions or other seasons. Fig. 5 shows the albedo seasonal relative trends for the 4 regions coincident with AQS in situ NO$_2$ data. The OMI DOMINO v2 incorporates a climatology albedo dataset (Kleipool et al., 2008) with snow/ice albedo adjustment using the NASA Near-real-time Ice and Snow Extent (NISE) dataset (Boersma et al., 2011). The climatology albedo data exhibits no trends. Thus, the trends of albedo mainly originate from the yearly variation of NISE detected snow/ice, followed by OMI sampling variation. The noticeable seasonal trend of the OMI DOMINO v2 albedo dataset is the 3.9% yr$^{-1}$ increase in DJF of the Midwest and a smaller DJF increase (1.0%) of the Northeast. In contrast, the MODIS albedo dataset exhibits a smaller positive DJF trend (0.8% yr$^{-1}$), 3.1% yr$^{-1}$ less than the trend from DOMINO v2, in the Midwest, and a small negative DJF trend (-0.8%) in the Northeast. The comparison to the AQS data shows that the MODIS albedo update leads to better agreement between satellite and in suit trends in winter in these regions (Fig. 4).

### 3.1.3 Improvement due to lightning filter

As discussed in Section 2.3.3, lightning NOx affects the retrievals of satellite tropospheric NO$_2$ VCDs. Fig. 6 shows that the lightning filter significantly reduces the difference between the OMI-based relative trend and that of the AQS data by 0.5-1.4% yr$^{-1}$ in the Northeast and 0.9-1.3% yr$^{-1}$ in the South. As a result, the seasonal trend differences are within 0.9% yr$^{-1}$ in these two regions except during SON. The lightning filter has little effect on the West and the Midwest. While lightning NOx can be significant during the monsoon season in some regions of the West (Fig. 3), the average tropospheric NO$_2$ VCDs are usually $< 1 \times 10^{15}$ molecules cm$^{-2}$ and lightning affected regions are therefore excluded in trend analysis.

The effect of lightning filter (Fig. 6) cannot be shown in Fig. 4 because the coincident OMI and AQS data points are fewer after applying the lightning filter. We examine the improvements of ocean trend removal, MODIS albedo update, and lightning filter by comparing the differences of different OMI-based seasonal relative trends from the AQS in situ trends in Fig. 7. The previously discussed improvements such as OMI albedo update for the Midwest and the Northeast during DJF are shown. By subtracting the AQS trends, we can now find clear improvements of the lightning filter for the South and the Northeast. There remains seasonal variation of OMI-based trend biases relative to in situ data but the discrepancies of the annual trends after the three discussed procedures are relatively small at 0.3% yr$^{-1}$, -0.3% yr$^{-1}$, -0.1% yr$^{-1}$, and 0.0% yr$^{-1}$, in the West, the Midwest, the Northeast, and the South regions (Fig. 1), respectively. The remaining seasonal difference of the trends reflects in part the nonlinear photochemistry (Gu et al., 2013).



### 3.2 OMI-based NO₂ trends

Table 1 summarizes the regional annual trends of coincident AQS in situ and OMI data. The "Standard" OMI data (following the DOMINO v2 algorithm) tend to show less $NO_2$ reduction than AQS data. After applying the three corrections discussed in the previous section to the OMI data, the agreement with the AQS trends is within the uncertainties of the trends.

Without the lightning filter, AQS decreasing trends are stronger while the decreasing trends of OMI data are less (Fig. 7). The lightning trend in the NLDN data is unclear due in part to the changing instrument sensitivity (Koshak et al., 2015). If lightning NOx is not accounted for in OMI retrieval, tropospheric $NO_2$ VCDs are overestimated. On the other hand, lightning accompanies low pressure systems which mix the atmosphere vertically and tend to reduce surface $NO_2$ concentrations when anthropogenic emissions are high such as urban and suburban regions. Therefore, lightning has opposite effects on surface and

satellite trends. The low-pressure dilution effect on surface $NO_2$ concentrations depends on anthropogenic emissions (since the end point of dilution is the background $NO_2$ value). Therefore, the reduction of decreasing surface trends likely reflects a reduction of low-pressure dilution effect. Similarly, as anthropogenic emissions decrease, the positive bias of tropospheric VCDs due to lightning NOx becomes larger, likely resulting in a reduction of decreasing trends. We consider the lightning effects on surface $NO_2$ trends to be mostly meteorological driven not by lightning NOx directly (e.g., Ott et al., 2010; Lu et

al., 2017) and hence the corrected OMI $NO_2$ data are likely closer to emission related concentration changes.

The AQS in situ $NO_2$ annual relative trends (coincident with OMI data with lightning filter) are most significant in the Northeast (-5.2±0.6% yr⁻¹) and the West (-4.2±0.5% yr⁻¹), followed by the South (-3.0±0.5% yr⁻¹) and the Midwest (-2.8±0.6% yr⁻¹) regions. The nationwide annual trend is -4.1±0.4% yr⁻¹, which is consistent with the previous studies (Lamsal et al., 2015; Lu et al., 2015; Tong et al., 2015; de Foy et al., 2016b; Duncan et al., 2016; Krotkov et al., 2016). The significant $NO_2$

reductions result from updated technologies and strict regulations (Krotkov, et al., 2016). The corrected OMI-based $NO_2$ trends (coincident with AQS data) show similar reduction rates in the West (-3.8±0.4% yr⁻¹), the Midwest (-3.1±0.5% yr⁻¹), the Northeast (-5.3±0.7% yr⁻¹) and the South (-3.0±0.5% yr⁻¹) regions. The nationwide annual trend is -3.9±0.3% yr⁻¹.

One advantage of satellite observations over a surface monitoring network is spatial coverage. The OMI data ("Lightning filter") coincident with the AQS data show a national annual trend of -3.9±0.3% yr⁻¹ similar to the AQS in situ trend of -

4.1±0.4% yr⁻¹. Using all data available (Fig. 8, Table 1), the OMI data ("Lightning filter") show a much lower trend of -1.5±0.2% yr⁻¹, about half of the AQS trend (-3.9±0.4% yr⁻¹). Fig. 9 shows that the AQS sites, which are mostly urban and suburban sites, tend to be located in regions with high tropospheric $NO_2$ VCDs. The OMI decreasing trend with corrected data is a function of tropospheric $NO_2$ VCDs, increasing from 0% yr⁻¹ to -6% yr⁻¹ (Fig. 9). The national annual trend is close to the value of clean regions which contribute much more than polluted regions. The larger decrease near the anthropogenic source

regions reflect in part the nonlinear photochemistry (Gu et al., 2013) and in part to a stronger influence of NOx sources such as soils in rural regions.



## 4. Conclusions

Using data from the DOMINO v2 algorithm, we find that the computed OMI-based seasonal $NO_2$ (relative) trends underestimate the decreasing trends of the EPA AQS data by up to 3.7% $yr^{-1}$. We attribute most of the discrepancies to OMI retrievals since the standard retrieval algorithm was not specifically designed for trend analysis. In this study, we show that

removing the background ocean trend (likely a result of the increasing stripes), adopting MODIS albedo data (with better temporospatial resolutions), and excluding lightning influences can bring OMI tropospheric $NO_2$ VCD trends in close agreement (within 0.3% $yr^{-1}$) with those of the AQS data. The largest effects of MODIS albedo update are in winter in Midwest and Northeast and those of lightning filter are in the South and the Northeast.

The national annual trend of the corrected OMI data is -1.5±0.2% $yr^{-1}$, about half of the AQS trend (-3.9±0.4% $yr^{-1}$). It

reflects that the AQS sites are mostly located in the urban and suburban regions, where OMI data show much larger decreasing trends (up to -6% $yr^{-1}$) than rural regions (down to 0% $yr^{-1}$). The reasons for the dependence of OMI derived trends on tropospheric $NO_2$ VCDs and the seasonal/regional trend differences are still not completely understood. Further studies are necessary to improve our understanding of these trends. The observation-based lightning filter implemented in this study is preliminary. Incorporating chemical transport modeling may improve this filter. Moreover, the results presented here represent

an alternative and indirect way to assess the importance of lightning NOx for National Climate Assessment (NCA) analyses described in Koshak et al. (2015), and Koshak (2017). Inversion studies (e.g., Zhao and Wang, 2009; Gu et al., 2013, 2014, 2016) will be needed to understand the emission changes corresponding to the OMI tropospheric $NO_2$ VCD trends.

**Acknowledgements**

This work was supported by the NASA ACMAP Program and the NASA Climate Indicators and Data Products for Future National Climate Assessments (NNH14ZDA001N-INCA). We thank data sources, including DOMINO v2 OMI data from KNMI, MODIS data from NASA, and EPA AQS $NO_2$ data from EPA. In addition, the authors gratefully acknowledge Vaisala Inc. for providing the NLDN data used in this study. K. Folkert Boersma acknowledges funding from the EU FP7 project QA4ECV (grant no. 607405).

**Data access**

The datasets used in this research have been obtained online as follows:

- DOMINO v2 $NO_2$ retrievals: http://www.temis.nl/airpollution/no2.html
- EPA AQS $NO_2$ data: US Environmental Protection Agency. Air Quality System Data Mart [internet database] available at http://www.epa.gov/ttn/airs/aqsdatamart.
- NLDN lightning data: https://lightning.nsstc.nasa.gov/data/data_nldn.html
- MODIS MCD43B3 data: https://lpdaac.usgs.gov/dataset_discovery/modis/modis_products_table/mcd43b3



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





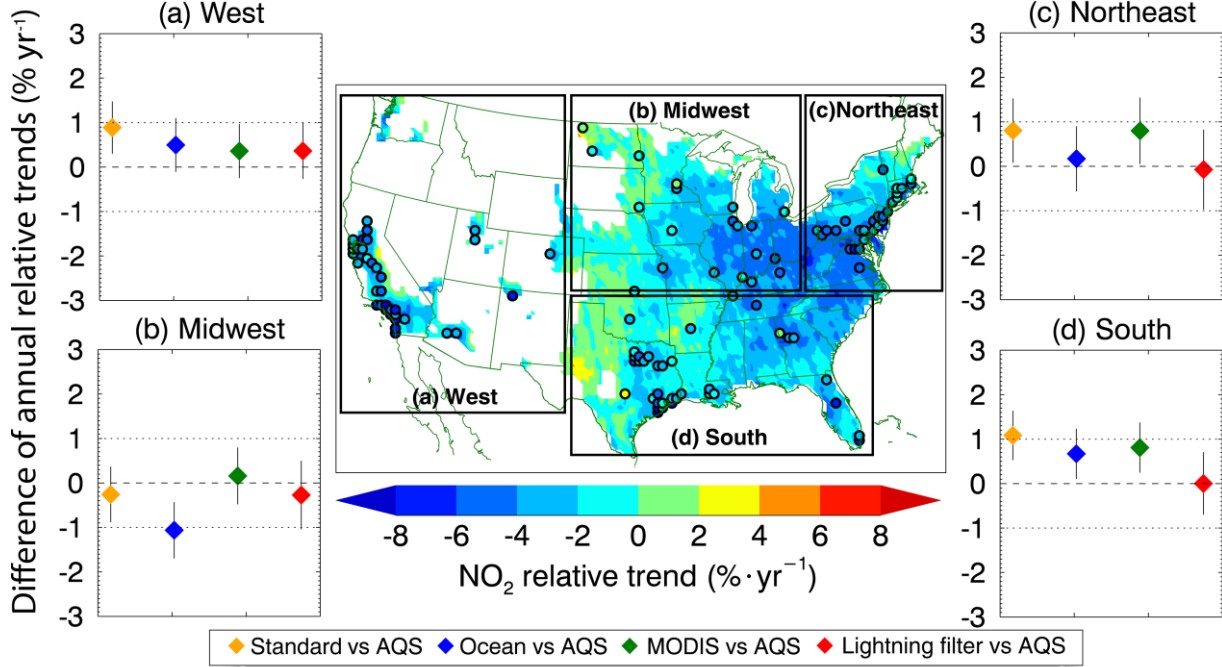

**Figure 1. The solid black borders in the center map define the four regions used in this study. The colored background shows the OMI-based NO₂ annual relative trends of the "lightning filter" data. Grid cells with 2005-2014 mean NO₂ VCD values < 1x10$^{15}$ molecules cm$^{-2}$ are excluded in this study and are shown in white. The black bordered circles represent the locations of AQS sites. Panel (a) through (d) show the regional difference of annual relative trends between coincident OMI-based and AQS in situ data. The colored diamonds are for "Standard" (orange), "Ocean" (blue), "MODIS" (green), and "Lightning filter" (red) OMI data, respectively. The different OMI VCD data are described in Section 2.4.**





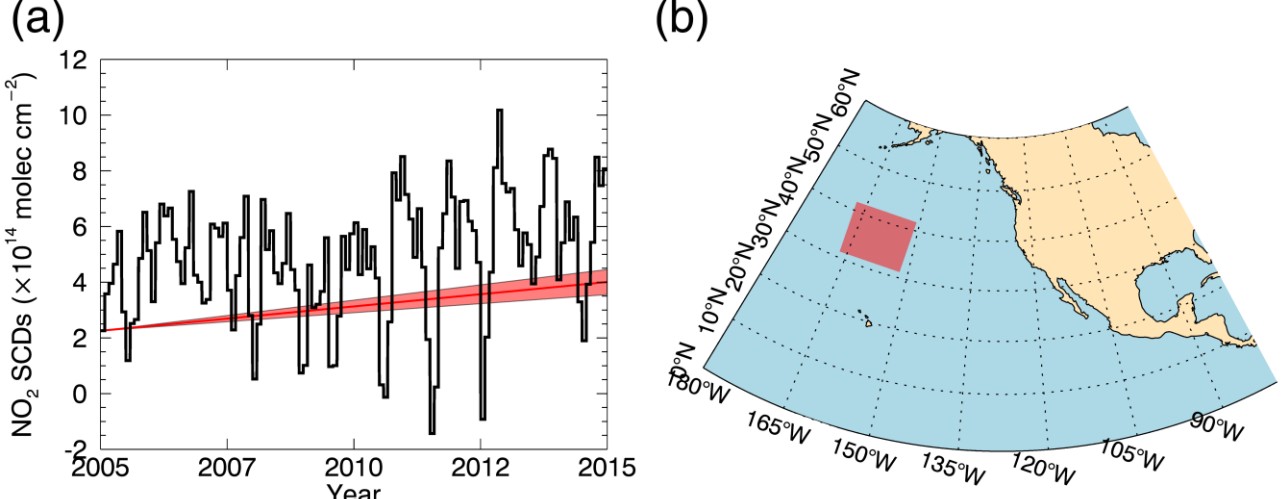

**Figure 2. The black line in panel (a) shows the monthly averaged OMI tropospheric NO₂ VCD values in the North Pacific region (red box in panel (b)) from 2005 to 2014. The red line in panel (a) represents the ocean trend used in this research, with the 95% confidence intervals shaded in red.**

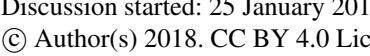



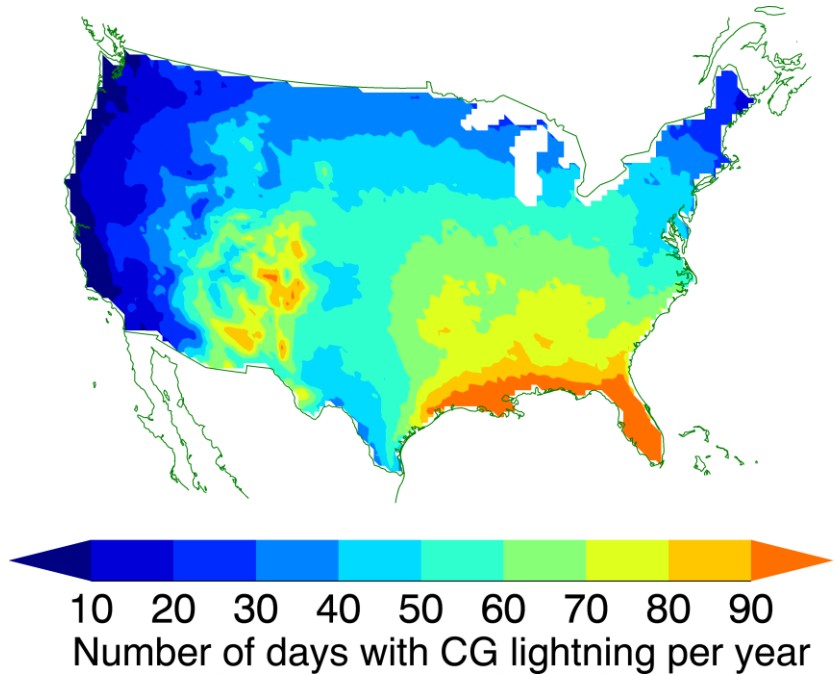

**Figure 3. Number of days with NLDN detected CG lightning per year during 2005-2014. The lightning occurrences are calculated using the REAM grid resolution.**




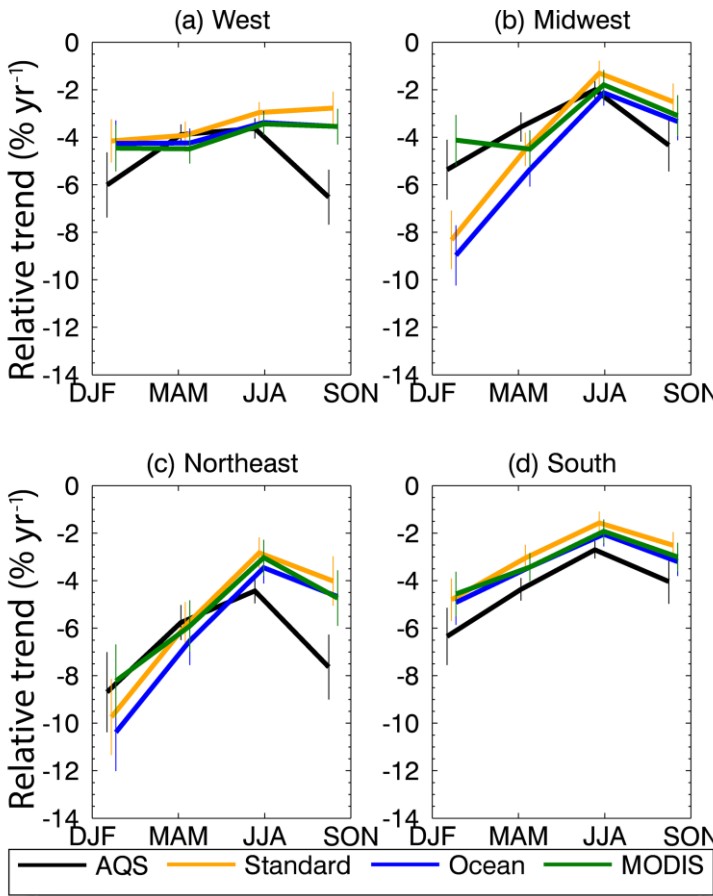

**Figure 4. Seasonal relative trends of NO₂ calculated from the AQS in situ measurements ("AQS", black line) and those derived from different OMI VCD data ("Standard", orange line; "Ocean", blue line; "MODIS", green line). The error bars represent 95% confidence intervals.**



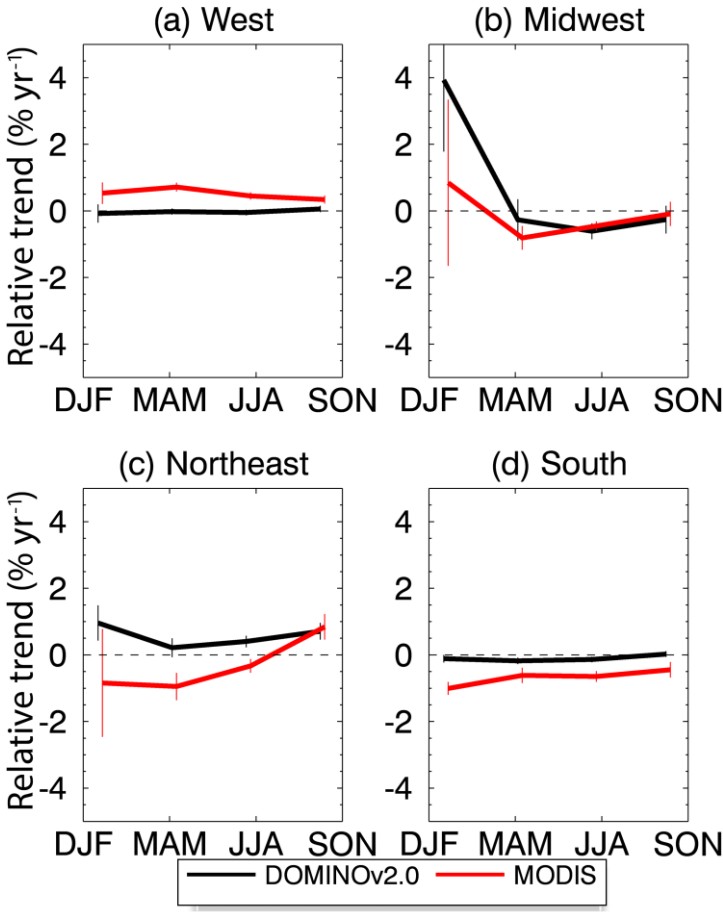

**Figure 5. Seasonal relative albedo trends of OMI (black line) and MODIS (red line) surface reflectance products, coincident with AQS in situ data used in Figure 5. The error bars represent 95% confidence intervals.**



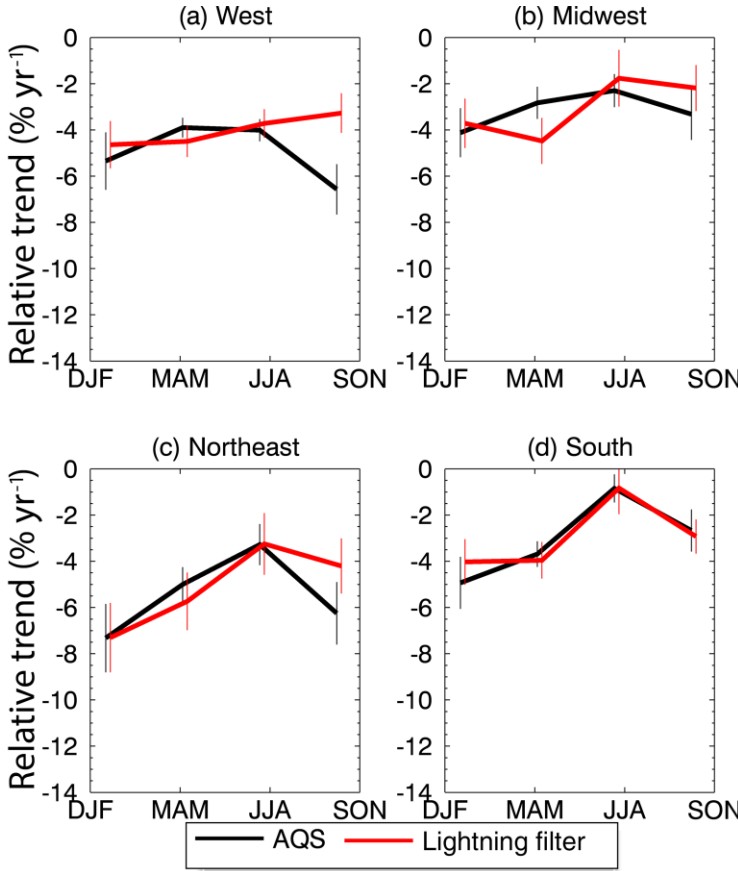

**Figure 6.** Same as Figure 5 but for coincident AQS (black line) and OMI data (red line) after applying the lightning filter. The coincident data points are less than used in Figure 5 and therefore the AQS trends are not the same.





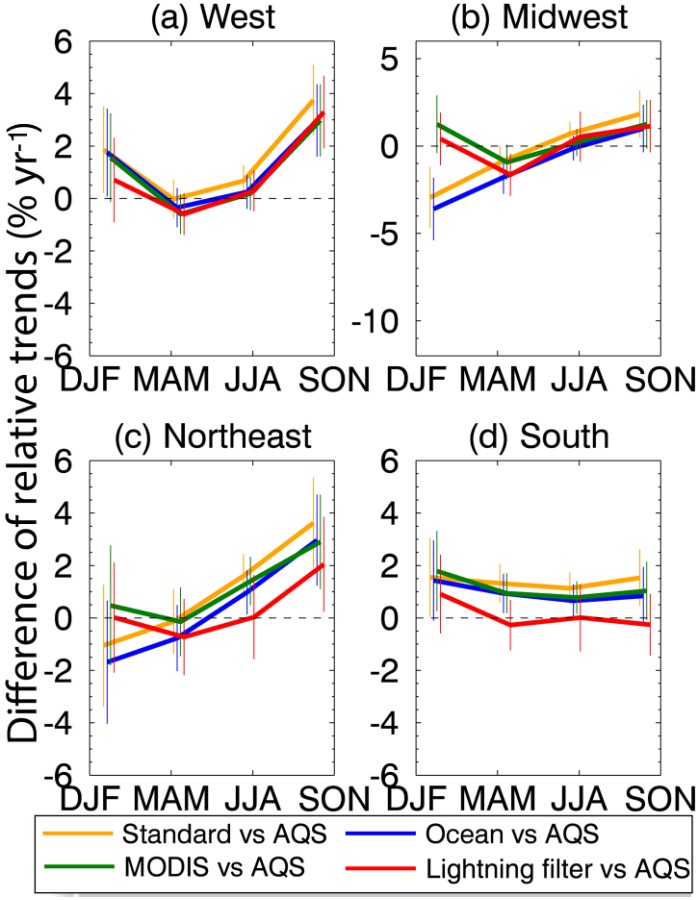

**Figure 7. Seasonal differences of OMI-based relative trends from those computed from AQS in situ data. The relative trends are shown in Figs. 6 and 8. The figure legends are the same as in Figs. 6 and 8 but with the AQS trends subtracted from the OMI-based trends.**



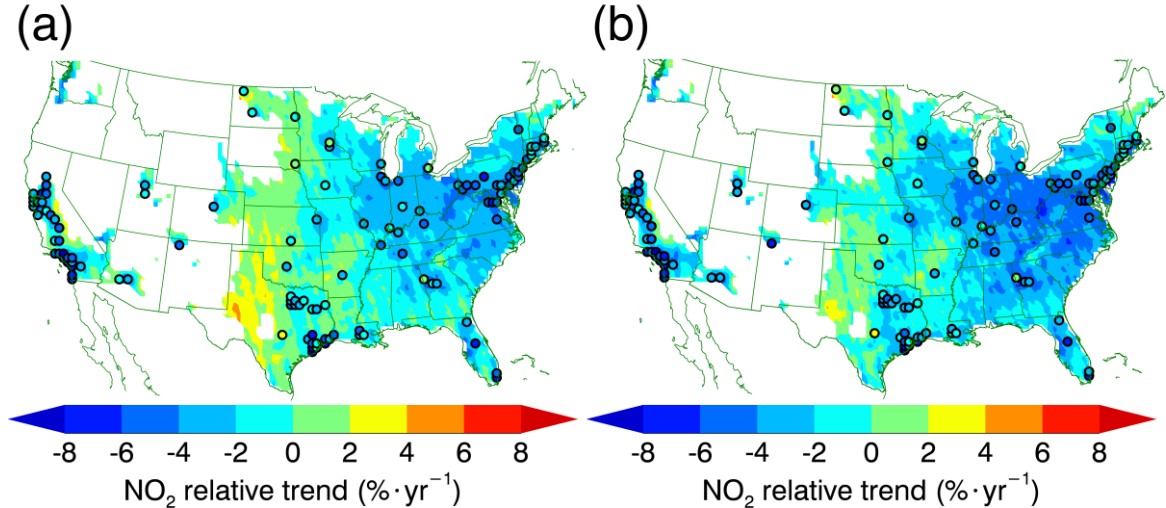

**Figure 8:** Annual relative trends of OMI-based NO₂ for "Standard" (a) and for "Lightning filter" (b) as the colored background. Black bordered circles indicate corresponding AQS NO₂ trends. Grid cells with 2005-2014 mean NO₂ VCDs < 1x10¹⁵ molecules cm⁻² are excluded in the analysis and are shown in white.





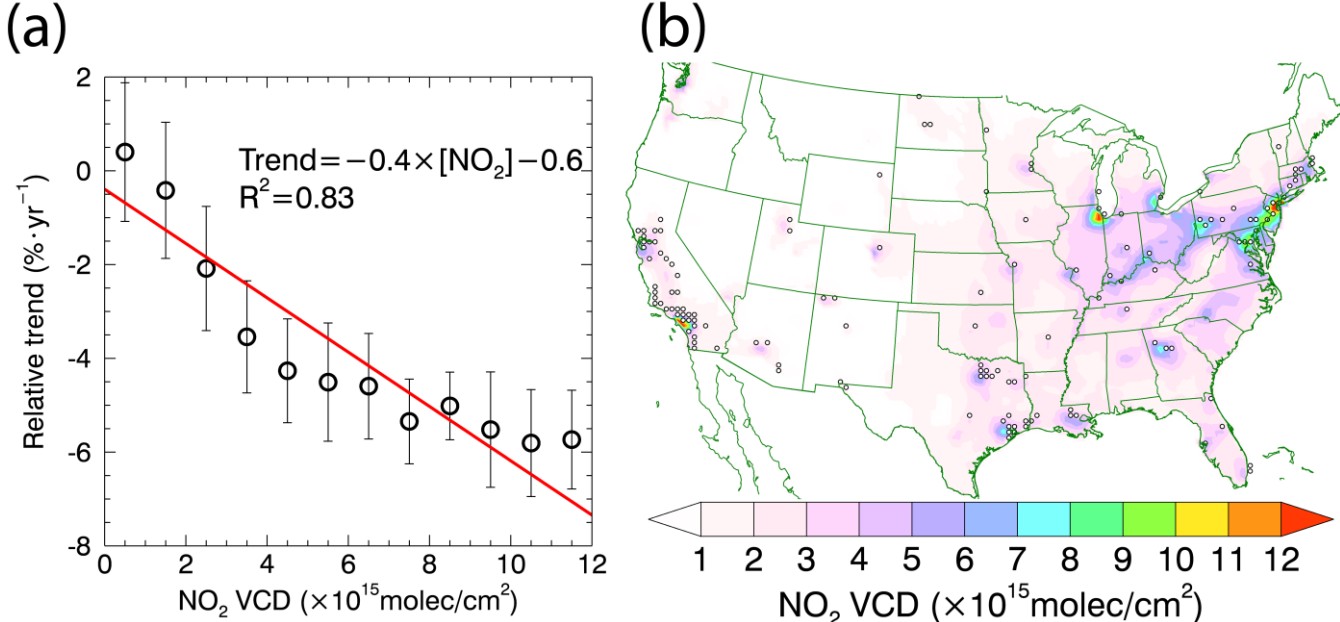

**Figure 9. (a) "Lightning filter" OMI-based NO₂ relative trend as a function 2005-2014 averaged OMI tropospheric NO₂ VCD binned every $1 \times 10^{15} molec/cm^2$. Red line shows a least-squares regression. (b) The distribution of 2005-2014 averaged OMI tropospheric NO₂ VCD. Black bordered circles represent AQS sites. The corrected OMI tropospheric NO₂ data are used.**





**Table 1. Annual relative trends calculated with coincident data and all available data. 95% confidence intervals from Mann-Kendall method are also listed.**

| Region | Annual relative trends of coincident data (% yr$^{-1}$) | | | | Annual relative trends using all data (% yr$^{-1}$) | | | |
|---|---|---|---|---|---|---|---|---|
| | Standard | | Lightning filter[a] | | Standard | | Lightning filter | |
| | AQS | OMI | AQS | OMI | AQS | OMI[b] | AQS | OMI[b] |
| West | -4.1±0.5 | -3.2±0.4 | -4.2±0.5 | -3.8±0.4 | -4.1±0.5 | -0.9±0.4 | -4.2±0.5 | -2.0±0.3 |
| Midwest | -3.4±0.5 | -3.6±0.4 | -2.8±0.6 | -3.1±0.5 | -2.5±0.5 | -0.9±0.4 | -2.2±0.5 | -1.8±0.4 |
| Northeast | -5.8±0.5 | -5.0±0.5 | -5.2±0.6 | -5.3±0.7 | -4.7±0.5 | -3.0±0.4 | -4.1±0.5 | -3.1±0.5 |
| South | -3.8±0.4 | -2.7±0.3 | -3.0±0.5 | -3.0±0.5 | -3.5±0.4 | -0.2±0.4 | -3.0±0.5 | -0.9±0.3 |
| Nationwide | -4.3±0.4 | -3.5±0.3 | -4.1±0.4 | -3.9±0.3 | -4.0±0.4 | -0.7±0.3 | -3.9±0.4 | -1.5±0.2 |

[a] These data include the three corrections of this study, namely, ocean trend correction, MODIS albedo update, and lighting filter screening.

5    [b] The spatial coverage is shown in Figure 1.