# Peer review of "Comparing OMI-based and EPA AQS in situ NO2 trends: Towards understanding surface NOx emission changes"

_Atmospheric Measurement Techniques, 2017_

## Short Comment (SC1) · 2 Feb 2018

Hi Ruixiong,

Interesting paper, I have a couple of comments. Checking for trends in OMI data can be tricky, especially due to the row anomaly.

One point I would like to raise is whether or not the observed trends you find are (1) actually 'real' - or in other words not caused by trends in the cloud fraction, cloud-top pressure, the AMFs or number of samples, and (2) statistically significant. Both aspects are important when comparing to real data. You might be getting better agreement to the trends in the in-situ data - but is that because the OMI NO2 is improved with your corrections or because a trend, in say the AMFs, has been introduced? I honestly don't

know but you should at least consider it.

Please see the discussion in my paper:

Barkley, M. P., González Abad, G., Kurosu, T. P., Spurr, R., Torbatian, S., and Lerot, C.: OMI air-quality monitoring over the Middle East, Atmos. Chem. Phys., 17, 4687-4709, https://doi.org/10.5194/acp-17-4687-2017, 2017.

Section 2.3.1 Ocean trend. Is the trend statistically significant? In my paper we looked at OMI NO2 over the Pacific (60 N–60 S, 90–170 W) - we couldn't find a statistically significant trend - but hey we maybe gridded the data in a different way!

Section 2.3.3 The lightning filter. I like this approach but aren't you introducing a sampling trend by losing these observations? I assume this is done per OMI observation - in which case you changing the footprint resolution too.

Anyways, good luck with the paper.

---

## Author Comment (AC1) · 6 Mar 2018

**Response:**

We thank Dr. Barkley for his comments and the references. The issues of AMF trends and statistical significance were discussed in the manuscript. The authors' responses are in **bold**.

*One point I would like to raise is whether or not the observed trends you find are (1) actually 'real' - or in other words not caused by trends in the cloud fraction, cloud-top pressure, the AMFs or number of samples, and (2) statistically significant. Both aspects are important when comparing to real data. You might be getting better agreement to the trends in the in-situ data - but is that because the OMI NO2 is improved with your corrections or because a trend, in say the AMFs, has been introduced?*

**The surface albedo update is aimed at reducing false signals in AMF. As stated in Section 3, we constructed the uncertainties of the trends with a confidence level of 95%, thus the derived trends are statistically significant. OMI-based trends are compared with spatially and temporally coincident EPA in situ trends. By selecting coincident in situ and OMI data, the effects of varying samples are already accounted for. Improving surface albedo data using MODIS products reduces the discrepancies between OMI-based and EPA in situ NO₂ trends in the Midwest and the Northeast. There are good physical reasons for the improvements, i.e. the variation in snow albedo. In essence, slant column trends can potentially be affected by the factors you mentioned. Properly processed vertical columns minimize their effects and thereby allow for comparison with in situ observations.**

*Section 2.3.1 Ocean trend. Is the trend statistically significant? In my paper we looked at OMI NO2 over the Pacific (60 N–60 S, 90–170 W) - we couldn't find a statistically significant trend - but hey we maybe gridded the data in a different way!*

**We calculated the ocean trend using the Slant Column Densities (SCDs) at the northern latitudes similar as the continental United States, which is the region of interest in our analysis. We used SCDs instead of Vertical Column Densities (VCDs) since the sampling stripes directly affect SCDs. Please note that the ocean trend is indeed small with a magnitude of $10^{13} \, molecues \, cm^{-2} \, yr^{-1}$. The different selections of regions (Northern Pacific vs Pacific), NO₂ products (SCDs vs VCDs), any difference in trend analysis methodology (such as accounting for seasonal variations in the trend analysis in the Mann-Kendall method we used) can all contribute to the discrepancies between observed ocean trends.**

*Section 2.3.3 The lightning filter. I like this approach but aren't you introducing a sampling trend by losing these observations? I assume this is done per OMI observation in which case you changing the footprint resolution too.*

**The lightning filter will decrease the data availability from 2%-27% as stated in Section 2.3.3. In places of significant lightning trends, this may introduce a sampling bias. However, the EPA data are also filtered out such that only coincident data were used in the comparison. Given the large effects of lightning NOx on OMI observations, we suggest that the lighting filter is essential for satellite-based trend analysis.**

---

## Referee Comment (RC1) · Anonymous Referee #1 · 24 Mar 2018

Review of "Reconciling the differences between OMI-based and EPA AQS in situ NO$_2$ trends" by Zhang et al.

This manuscript summarizes the sensitivity of OMI NO$_2$ trend to several factors such as a baseline trend (over the ocean), surface albedo, and lightning filter. I found the information in the manuscript is useful. The paper is well organized and presentations are neat. But I think that general conclusions (or contents in the abstract) are misleading and some important analyses are missing. I suggest to revise the manuscript before final publication based on the comments below.

First, I do not agree with the authors in the abstract line 14-15 ("how to improve OMI NO$_2$ retrievals for more reliable trend analysis") and line 23-25 ("we recommend future studies to apply these procedures to ensure the quality of satellite based NO$_2$ trend analysis, especially in regions without reliable long-term in situ NO$_2$ measurements"). I think the agreement between the trends in surface monitored data (AQS) and those in standard OMI (Table 1) is already good, considering uncertainties in the satellite retrievals and the spatial coverage of the surface monitors. The authors need to clarify the spatial resolution of the OMI data (used in the trend analysis) and the spatial extent (or representativeness) of the ground-based observations. It is exciting to see better agreement between the trends from AQS and the final OMI (lightning filter) in Table 1. But I am not sure that these two should agree exactly. Figure 4 shows that the effects of different OMI retrievals are not clear except DJF in Midwest and Northeast. If summertime or typical ozone season satellite data are used for the trend study, it is not worth trying different retrievals or corrections suggested in this study.

In general, the manuscript reports the impact of uncertainties of satellite tropospheric NO$_2$ retrievals on the trend analysis. This is a sensitivity test study that provides useful information and can be a good reference to summarize the uncertainties in the OMI NO2 based trend analysis. However, I do not think it has broad and substantial impacts to change or shape future research.

One thing missing is a validation of *a priori* model NO$_2$ profile or near surface NO$_2$. According to this study, NO$_2$ columns (potentially NO$_x$ emissions) decrease by ~40% for 10 years. How does satellite NO$_2$ column retrieval change if *a priori* profiles come from the model results incorporating this reduced NO$_x$ emission (e.g., 40% reduction to 70% reduction considering a potential error in the emission).

Final comment is to elaborate the correction of NO$_2$ measurements by molybdenum converter. The plot in supplementary material (Figure S1) can include more details for the 7 sites. The diurnal variation in each season (if not month) and standard deviation in the plot will be helpful to characterize the ratios between surface NO$_2$ concentrations of chemiluminescence to photolytic instruments. The plots for each site (7 sites) will be useful for readers.

---

## Referee Comment (RC2) · Anonymous Referee #2 · 31 Mar 2018

The manuscript *Reconciling the differences between OMI-based and EPA AQS in situ NO2 trends* by Zhang et al. is an investigation of the differences between trends in tropospheric NO$_2$ columns derived from the OMI satellite instrument and those derived from the EPA AQS network. This is an important and interesting research question, as in remote areas one often has to rely on remote sensing data in order to get reliable measurements of air quality. The manuscript falls well within the scope of AMT.

That being said, the manuscript fails to convince the reader regarding the comparability of the two datasets to begin with. Also, the manuscript is often too imprecise.

Most of the following points are minor and can be fixed by providing more precise information about what the authors did exactly, but they should be addressed before publication in AMT:

[Figure]

**1 Comparing VCDs and surface concentrations**

The authors fail to convince the reader why OMI VCDs, which are the integrated $NO_2$ content of the troposphere at a given location, should be comparable to the in-situ surface concentrations of the AQS dataset. There have been numerous studies trying to relate the two measures to each other, and it should be clear to the authors that in order to compare the two, one has to take special caution. This becomes most problematic in the discussion of the effect of the lightning filter, where the authors leave the impression that lightning leads to "wrong" OMI VCDs.

In a revised manuscript, the authors should include a summary of the problems arising from comparing the integrated satellite to the in-situ point measurements, shold reference relevant literature, and should make sure that they consider these differences in the comparisons of the relative trends. Also, they should explicitly discuss the problems arising from comparing *relative trends* of these two different measures.

**2 Definition of the relative trends**

- 07/05: It is not entirely clear how exactly the authors calculate the relative trends. Is it a linear trend, calculated by linear regression? By default, the Mann-Kendall test is non-parameteric. If the authors use the *Sen slope estimator* as *relative trend*, they should explicitly say so. Otherwise, the authors should explicitly say what the reference value is for the *relative* trends (i.e., 2005, or average of the whole period, or ...).

- In some places, the authors do give an uncertainty of relative trends. However, they do not give enough detail about how these trend uncertainties are being calculated. If they indeed use the *Sen slope estimator* from the Mann-Kendall test as relative trend, it is unclear how they define the uncertainty of this estimate.

This is however crucial in order to evalue if the improvements in the agreement of OMI and AQS relative trends are statistically significant at all. Furthermore, in some Figure captions the authors indicate *95% confidence intervals*; please briefly describe in the text how these are derived.

- Another point regarding the trend calculations is the uncertainties of the relative trends. The notion of *difference between OMI and AQS trends* only makes sense if there is some way of assessing if these differences are statistically significant at all.

**3  Importance of yearly varying NOx emissions**

05/04: The authors claim that the *yearly variations of [. . . ] anthropogenic emission changes have little impact on trend analysis results*, and they cite a paper by Lamsal et al. (2015). However, in the cited paper, Lamsal et al. state (Sect. 2.2.1):

In this work, we further improve the operational OMI NO2 retrieval [. . . ] by using new a priori NO2 profiles [. . . ] with year-specific emissions. The year-specific emissions not only improve the representation of the NO2 vertical distribution, but also capture the yearly changes in NO2 profile shapes. The latter is critical due to rapid decline in the U.S. NOx emissions in recent years [. . . ].

Since the present study deals with the time period 2005–2014, I do not see how the authors' choice to use fixed 2010 NOx emissions is backed by the cited work by Lamsal et al. Given the fact that the study period does include the years of economic crisis, the authors' choice to use fixed emissions is questionable. I strongly suggest some quantitative assessment of the influence of using fixed emissions.

**4 Reconciling chemiluminescent and photolytic in-situ measurements**

The authors claim that calculating a correction factor for the chemiluminescent in-situ data by taking the average ratio of chemiluminescent to photoltic measurements. This would only work properly if the reasons for the high bias of the former instruments were identical at all measurment stations. While it is true that this correction does not influence the relative trends, the authors should at least mention this.

**5 Importance of individual sources of AMF uncertainty**

04/28: The authors claim that *the first two factors* are most important for the $NO_2$ trend analysis, but fail to back up their claim.

**6 Time span of lightning filter**

06/20: The authors' choice of lightning filter (72hrs / 90km) seems arbitrary and needs to be justified. As the auhors correctly state, the lifetime of NOx in the free troposphere can reach up to one week. By making their filter only 90km wide, a back-of-the-envelope calculation quickly shows that the NOx produced by a single lightning occurrence can easily be transported considerably further within 72hrs than only 90km. The authors seem to be aware of this inconsistency, because they introduce an additional filter for the Northeast which depends on lightning occurrence in the South, implying a transport distance of many hundreds of kilometers.

**7 Minor comments**

- 04/18: NO$_2$ *partial* VCDs

- 05/06: Which trends? Those with the *default* albedo, or those with the *update*? . . . )

- 05/09: I personally find the name *ocean trend* misleading, as it has nothing to do with the ocean (except for the geographical location of the clean background region). Maybe the authors can come up with a name that somehow indicates the origin of the trend (e.g., *instrument drift*).

- 07/10: It seem that there are *four* different OMI-based NO$_2$ trends

- 07/15: To avoid confusion, please explicitly mention that these are *absolute* differences of the *relative* trends.

- 09/05: *trends of OMI data are less* — than what?

- 09/07: OMI VCDs are *not* overestimated when not filtering for lightning NOx - the lightning NOx is part of the VCD. It leads to worse agreement between OMI and AQS trends, but then again, these are two fundamentally different measures anyways.

- 09/11: What is a *reduction of decreasing surface trends*? Misleading phrase, since the trends are decreasing trends to begin with. Maybe it'd be better to say *stronger decreasing trends* or something similar.

- 09/12: Again, OMI VCDs are not *biased* due to lightning, see above.

- 09/13: *reduction of decreasing trends* — see above

- 09/15: OMI VCDs are not wrong when they include lightning NOx – the authors should therefore not make the qualitative statement *corrected* here. *Filtered* would be better.

- 09/29: I would assume that the driving factor in stronger decreasing trends close to anthropogenic source regions is the decreasing emissions in those, resulting in less trensported NOx in those areas.

- 10/03: Since comparing VCDs to surface concentrations is a difficult issue to begin with, I would not blame the OMI retrievals for the differences – when comparing apples and oranges, why should one blame one and not the other for the differences? Saying that the OMI data are *not designed for trend analysis* doesn't make sense. If one has to *design* a dataset in order to be able to do trend analysis, maybe there just are no significant trends in the underlying data to begin with?

- In Fig. 1a-d, it is not clear if positive numbers mean that the OMI trend or the AQS trend is higher. Please update the Figure caption with a mathematically precise description (e.g., "OMI relative trend minus AQS relative trend").

- Fig. 3: Please update the Figure caption with a *precise* indication of the units, e.g., "number of days [. . . ] per REAM grid cell". Also, please spell out *cloud-to-ground* instead of *CG* in the caption.

- Fig. 6: Please indicate $NO_2$ somewhere in the Figure caption. Also, the legend for the OMI data should be something like *OMI (lightning filter)*; after all, the data show trends of OMI $NO_2$ columns and not of *the lightning filter*.

- Fig. 7: I don't understand what *the figure legends are the same as in Figs. 6 and 8* is supposed to mean. Please clarify.

- Fig. 7: Please explicitly indicate in the Figure caption if statistically insignificant trends are shown or not.

- Fig. 9a: There is something wrong with the Figure caption, it does not contain a complete sentence (maybe there's just a *of* missing?). Please indicate what the barbs on the individual data points mean.

- Fig. 9b: Please be specific about which OMI $NO_2$ data you show in this Figure, using the nomenclature from earlier. As explained above, the notion of *corrected* is misleading.

---

## Author Comment (AC2) · 26 Apr 2018

We would like to thank Referee #1 for the insightful comments. We revised the manuscript accordingly. We present our responses and changes below. The reviewer's comments and suggestions are in *italic*. Authors' responses are in **bold**.

**Response to Referee #1:**

*This manuscript summarizes the sensitivity of OMI $NO_2$ trend to several factors such as a baseline trend (over the ocean), surface albedo, and lightning filter. I found the information in the manuscript is useful. The paper is well organized and presentations are neat. But I think that general conclusions (or contents in the abstract) are misleading and some important analyses are missing. I suggest to revise the manuscript before final publication based on the comments below.*

**This research focuses on improving OMI $NO_2$ retrieval (instead of sensitivities) in trend analysis by removing the ocean trend (due to increasing stripes), using MODerate-resolution Imaging Spectroradiometer (MODIS) albedo in deriving air mass factors (AMFs), and applying the lightning flash filter. The three corrections are necessary for $NO_2$ trend analysis using OMI $NO_2$ retrieval.**

*First, I do not agree with the authors in the abstract line 14-15 ("how to improve OMI NO2 retrievals for more reliable trend analysis") and line 23-25 ("we recommend future studies to apply these procedures to ensure the quality of satellite based NO2 trend analysis, especially in regions without reliable long-term in situ NO2 measurements"). I think the agreement between the trends in surface monitored data (AQS) and those in standard OMI (Table 1) is already good, considering uncertainties in the satellite retrievals and the spatial coverage of the surface monitors. The authors need to clarify the spatial resolution of the OMI data (used in the trend analysis) and the spatial extent (or representativeness) of the ground-based observations.*

**We added in the abstract "**However, the current OMI tropospheric $NO_2$ retrievals are not designed for analyzing multi-year tropospheric $NO_2$ trends.**" It is not surprising that satellite data need to be reprocessed in trend analysis, which is true for all observation-based trend studies.**

**The reason Table 1 results may be conceived to be "already good" (at for coincident data) is because we showed only annual trends. It did show that for all data, the effects of our recommended data processing are > a factor of 2 for West, Midwest, and South (comparing the OMI column with and without the lightning filter). We now emphasized this point in the abstract and conclusions.**

**We further make the points that (1) processed OMI data show smaller trends (Figure 8) and (2) using surface data alone will have a tendency to overestimate $NO_2$ trends (Figure 9). Lastly, Figs. 4-7 show that on a seasonal basis, the effects of the data processing we recommended can be much larger than the annual mean changes of Table 1.**

**To deal with the surface measurement representativeness, we only compare to regional trends from surface observations, which are much more statistically representative. Previous studies did the same. As discussed above, we show the effects of site locations on the trend analysis in Fig. 10 (and Table 1). This point has not been emphasized previously and needs to be accounted for in future studies.**

*It is exciting to see better agreement between the trends from AQS and the final OMI (lightning filter) in Table 1. But I am not sure that these two should agree exactly.*

**As our research focuses on the regional relative trend difference, we expect that the surface $NO_2$ trends and $NO_2$ VCD trends should be very close (They are both affected by chemical non-linearity and may show trends different from emission trends). We stated this in Section 3.1, "**The $NO_2$ relative trends from

both datasets are expected to be close on a regional basis where surface emissions of NOx dominate the observed surface concentrations and tropospheric VCDs of $NO_2$." **As the reviewer already pointed out that the trends between surface and OMI coincident data are already in reasonably good agreement for annual trends, which is a reflection of this reasoning. We showed in this paper what improvements can be made further, the physical reasons, and the implications for understanding regional $NO_2$ trends.**

*Figure 4 shows that the effects of different OMI retrievals are not clear except DJF in Midwest and Northeast. If summertime or typical ozone season satellite data are used for the trend study, it is not worth trying different retrievals or corrections suggested in this study.*

**We do not quite understand this comment. Figure 4 only shows the effects of adding ocean and MODIS corrections only, which can be quite large for some regions and some seasons already. These effects obviously need to be understood. Figure 7 shows the effects of adding lightning filter to the previous two corrections. We think that it shows clearly that all three corrections are necessary for doing trend analysis.**

*In general, the manuscript reports the impact of uncertainties of satellite tropospheric NO2 retrievals on the trend analysis. This is a sensitivity test study that provides useful information and can be a good reference to summarize the uncertainties in the OMI NO2 based trend analysis. However, I do not think it has broad and substantial impacts to change or shape future research.*

*One thing missing is a validation of a priori model NO2 profile or near surface NO2. According to this study, NO2 columns (potentially NOx emissions) decrease by ~40% for 10 years. How does satellite NO2 column retrieval change if a priori profiles come from the model results incorporating this reduced NOx emission (e.g., 40% reduction to 70% reduction considering a potential error in the emission).*

**We need to clarify that this work is not a sensitivity test study but proposing three necessary corrections to satellite retrievals used in trend analysis. Due to the chemical non-linearity (Gu et al., 2016), the $NO_2$ VCD trend differs from NOx emission trend. As $NO_2$ columns decrease by ~40%, the NOx emissions decrease by less than ~40%.**

**Lamsal et al. (2015) reported that using 2005 emission profile results in an underestimation of generally less than 2% of overall 2005-2010 $NO_2$ reduction in polluted regions and that the trend is less sensitive to the vertical profile assumption in highly polluted areas (Section 2.2.2 and Figure 3, Lamsal et al., 2015). This is about an underestimation of about 0.3% $yr^{-1}$, which could be used to explain the residual discrepancy (0.0-0.4% $yr^{-1}$) between OMI-based and in situ $NO_2$ trends.**

**The proper way to estimate the effects of NOx emissions on $NO_2$ VCDs is to perform proper inverse modeling of NOx emissions (e.g., Gu et al., 2014). However, the satellite data still need to be processed such that factors we discussed in this work do not introduce artifacts in tropospheric $NO_2$ column trends. In the inverse modeling by Gu et al. (2014), the effects of AMF changes on inverse modeling will be accounted for, which we believe is the proper method for deriving NOx emission trends. That is to say that all proper inverse modeling studies of NOx emission trend should include AMF calculation that is derived from the a posteriori (not a priori) NOx emissions and it is the appropriate method of quantify the relative small effects. We added this point to the last sentence in the conclusions.**

**We revised the sentence in Section 2.3, as follows:**

"The yearly variations of meteorology and anthropogenic emission changes have little impact in polluted areas on trend analysis results using OMI data (Lamsal et al., 2015)."

**We revised one sentence in Section 3.1.3, as follows:**

"The remaining seasonal difference of the trends reflects in part the nonlinear photochemistry (Gu et al., 2013) and the effects of NOx emission changes on $NO_2$ retrievals (Lamsal et al., 2015)."

*Final comment is to elaborate the correction of NO2 measurements by molybdenum converter. The plot in supplementary material (Figure S1) can include more details for the 7 sites. The diurnal variation in each season (if not month) and standard deviation in the plot will be helpful to characterize the ratios between surface NO2 concentrations of chemiluminescence to photolytic instruments. The plots for each site (7 sites) will be useful for readers.*

**The diurnal variation of the ratio is irrelevant to this research, as OMI overpasses only at around 13:30 local time. Due to the lack of photolytic instruments in each site, we used the averaged ratio between chemiluminescent to photolytic $NO_2$ measurements in 7 available sites. The individual plots for these 7 sites can be misleading due to data availability. As we stated in Section 2.1, this correction will affect absolute trends but not relative trends, which are what we analyzed in this study. We added the 95th percentile confidence intervals to Figure S1.**

[revised manuscript text omitted]

---

## Author Comment (AC3) · 26 Apr 2018

We appreciate Referee #2 for the thoughtful and detailed comments. We revised the manuscript accordingly. We present our responses and changes below. Reviewers' comments and suggestions are in *italic*. Authors' responses are in **bold**.

**Response to Referee #2:**

*The manuscript Reconciling the differences between OMI-based and EPA AQS in situ NO2 trends by Zhang et al. is an investigation of the differences between trends in tropospheric NO2 columns derived from the OMI satellite instrument and those derived from the EPA AQS network. This is an important and interesting research question, as in remote areas one often has to rely on remote sensing data in order to get reliable measurements of air quality. The manuscript falls well within the scope of AMT.*

*That being said, the manuscript fails to convince the reader regarding the comparability of the two datasets to begin with. Also, the manuscript is often too imprecise.*

*Most of the following points are minor and can be fixed by providing more precise information about what the authors did exactly, but they should be addressed before publication in AMT:*

*1 Comparing VCDs and surface concentrations*

*The authors fail to convince the reader why OMI VCDs, which are the integrated NO2 content of the troposphere at a given location, should be comparable to the in-situ surface concentrations of the AQS dataset. There have been numerous studies trying to relate the two measures to each other, and it should be clear to the authors that in order to compare the two, one has to take special caution. This becomes most problematic in the discussion of the effect of the lightning filter, where the authors leave the impression that lightning leads to "wrong" OMI VCDs.*

**As our research focuses on the regional relative trend difference, we expect that the surface NO$_2$ trends and NO$_2$ VCD trends should be very close (They are both affected by chemical non-linearity and may show trends different from emission trends). We stated this in Section 3.1, "**The NO$_2$ relative trends from both datasets are expected to be close on a regional basis where surface emissions of NOx dominate the observed surface concentrations and tropospheric VCDs of NO$_2$.**" We showed in this paper what improvements can be made further, the physical reasons, and the implications for understanding regional NO$_2$ trends.**

**To deal with the surface measurement representativeness, we only compare to regional trends from surface observations, which are much more statistically representative. Previous studies did the same. As discussed above, we show the effects of site locations on the trend analysis in Fig. 10 (and Table 1). This point has not been emphasized previously and needs to be accounted for in future studies.**

**We added in the abstract** "However, the current OMI tropospheric NO$_2$ retrievals are not designed for analyzing multi-year tropospheric NO$_2$ trends." **It is not surprising that satellite data need to be reprocessed in trend analysis, which is true for all observation-based trend studies. We made strong points that standard OMI tropospheric NO$_2$ VCD data will introduce errors. We believe that our analysis results support these points.**

**Having said that, we understand the reviewer's point on lightning NOx. We clarified the discussion by adding** "While lightning NOx is part of OMI NO$_2$ observations, we treat the influence of lightning on the OMI tropospheric VCD trend as a bias for comparison purposes in this study since AQS data are not as strongly affected by lightning."

**We added in the conclusions** "While lightning NOx is part of OMI NO$_2$ observations, we treat the influence of lightning on the OMI tropospheric VCD trend as a bias for comparison purposes in this study since AQS data are not as strongly affected by lightning. Furthermore, lightning NOx effects need to be removed when

using satellite observations to understand the effects of changing anthropogenic emissions."

*In a revised manuscript, the authors should include a summary of the problems arising from comparing the integrated satellite to the in-situ point measurements, should reference relevant literature, and should make sure that they consider these differences in the comparisons of the relative trends. Also, they should explicitly discuss the problems arising from comparing relative trends of these two different measures.*

**Table 1 shows that even though the annual trends for coincident data can be conceived as "already good", the effects of our recommended data processing are > a factor of 2 for West, Midwest, and South (comparing the OMI column with and without the lightning filter). We now emphasize this point in the abstract and conclusions. We did not add criticisms of any specific previous study because it is inappropriate to speculate if "good" agreement between in situ and OMI trends for coincident data is the reason that previous studies did not study the data processing procedures we recommended.**

**We further make the points that (1) processed OMI data show smaller trends (Figure 8) and (2) using surface data alone will have a tendency to overestimate NO₂ trends (Figure 9). Lastly, Figs. 4-7 show that on a seasonal basis, the effects of the data processing we recommended can be much larger than the annual mean changes of Table (1).**

**The lightning leads to inaccurate OMI retrieved NO₂ VCDs. Current models have difficulty simulating lightning NOx and low-pressure system meteorology correctly across different years (as stated in Section 2.3.3 and 3.2), which affects NO₂ vertical profiles and subsequently leads to inaccurate AMFs and NO₂ VCDs.**

**We updated the manuscript regarding potential factors contributing to the divergence between OMI-based and in situ NO₂ trends.**

"Lamsal et al. (2015) also found the divergence between the annual trends inferred from the two datasets, i.e. -4.8% yr-1 vs -3.7% yr-1 during 2005-2008, and -1.2% yr-1 vs -2.1% yr-1 during 2010-2013. There are several potential factors attributing to the discrepancies between trends from satellite and ground-based measurements: interferences by the oxidation products of NOx from the chemiluminescent instruments (Lamsal et al., 2008, 2014, 2015), the differences of sampling time between OMI (~13:30 local time) and AQS (hourly) measurements (Tong et al., 2015), a high sensitivity of NO₂ VCDs to high-altitude NO₂ in contrast to the high sensitivity of surface NO₂ concentrations to surface NOx emissions (Duncan et al., 2013; Lamsal et al., 2015), spatial representativeness of satellite pixels (Lamsal et al., 2015), and high uncertainties of satellite retrievals in clean regions (Lamsal et al., 2015).

To understand how various factors and the retrieval procedure  affect the resulting OMI derived trends and their differences from those derived from the surface AQS measurements, we utilize a regional 3-D chemistry transport model (CTM), a radiative transfer model (RTM), and the Mann-Kendall method (Mann, 1945; Kendall, 1948) to calculate OMI-based NO₂ seasonal relative trends during Dec-Jan-Feb (DJF), Mar-Apr-May (MAM), Jun-Jul-Aug (JJA), and Sept-Oct-Nov (SON) (Section 2)."

*2 Definition of the relative trends*

*• 07/05: It is not entirely clear how exactly the authors calculate the relative trends. Is it a linear trend, calculated by linear regression? By default, the Mann-Kendall test is non-parameteric. If the authors use the Sen slope estimator as relative trend, they should explicitly say so. Otherwise, the authors should explicitly say what the reference value is for the relative trends (i.e., 2005, or average of the whole period, or . . . ).*

**Thank you. We are using Mann-Kendall method with the Sen's slope estimator. We now mention this**

**in the abstract and Section 3.**

"The Mann-Kendall method with the Sen's slope estimator is applied to derive the $NO_2$ seasonal and annual trends for four regions at coincident sites during 2005-2014."

"We apply the Mann-Kendall method with the Sen's slope estimator to calculate the relative trend of $NO_2$ for each season, i.e. DJF, MAM, JJA, and SON, during 2005-2014. We  compute the uncertainties of the trends with 95th percentile confidence intervals using the Mann-Kendall method ."

*• In some places, the authors do give an uncertainty of relative trends. However, they do not give enough detail about how these trend uncertainties are being calculated. If they indeed use the Sen slope estimator from the Mann-Kendall test as relative trend, it is unclear how they define the uncertainty of this estimate. This is however crucial in order to evalue if the improvements in the agreement of OMI and AQS relative trends are statistically significant at all. Furthermore, in some Figure captions the authors indicate 95% confidence intervals; please briefly describe in the text how these are derived.*

**We use Mann-Kendall method with the Sen's slope estimator to estimate the relative trends. The uncertainties are given as the 95th percentile confidence intervals. We now state this in Section 3:** "We compute the uncertainties of the trends with the 95th percentile confidence intervals using the Mann-Kendall method."

*• Another point regarding the trend calculations is the uncertainties of the relative trends. The notion of difference between OMI and AQS trends only makes sense if there is some way of assessing if these differences are statistically significant at all.*

**The uncertainties, the 95th percentile confidence intervals, are shown as the error bars in Figure 7.**

*3 Importance of yearly varying NOx emissions*

*05/04: The authors claim that the yearly variations of [. . . ] anthropogenic emission changes have little impact on trend analysis results, and they cite a paper by Lamsal et al. (2015). However, in the cited paper, Lamsal et al. state (Sect. 2.2.1):*

*In this work, we further improve the operational OMI NO2 retrieval [. . . ] by using new a priori NO2 profiles [. . . ] with year-specific emissions. The year-specific emissions not only improve the representation of the NO2 vertical distribution, but also capture the yearly changes in NO2 profile shapes. The latter is critical due to rapid decline in the U.S. NOx emissions in recent years [. . . ].*

*Since the present study deals with the time period 2005–2014, I do not see how the authors' choice to use fixed 2010 NOx emissions is backed by the cited work by Lamsal et al. Given the fact that the study period does include the years of economic crisis, the authors' choice to use fixed emissions is questionable. I strongly suggest some quantitative assessment of the influence of using fixed emissions.*

**Lamsal et al. (2015) reported that using 2005 emission profile results in an underestimation of generally less than 2% of the overall 2005-2010 $NO_2$ reduction in polluted regions and that the trend is less sensitive to the vertical profile assumption in highly polluted areas (Section 2.2.2 and Figure 3, Lamsal et al., 2015). This equals to an underestimation of about 0.3% yr$^{-1}$, which could be used to explain the residual discrepancy (0.0-0.4% yr$^{-1}$) between OMI-based and in situ $NO_2$ trends.**

**The proper way to estimate the effects of NOx emissions on NO₂ VCDs is to perform NOx emission daily retrieval inversion modeling (Gu et al., 2014) to derive daily NOx emissions. We added this point to the last sentence in the conclusions. However, obtaining more reliable satellite NO₂ retrievals (as in this study) is a prerequisite to such NOx emission inversion (for trend analysis).**

**We revised the sentence in Section 2.3, as follows:**

"The yearly variations of meteorology and anthropogenic emission changes have little impact in polluted areas on trend analysis results using OMI data (Lamsal et al., 2015)."

**We revised one sentence in Section 3.1.3, as follows:**

"The remaining seasonal difference of the trends reflects in part the nonlinear photochemistry (Gu et al., 2013) and the effects of NOx emission changes on NO₂ retrievals (Lamsal et al., 2015)."

*4 Reconciling chemiluminescent and photolytic in-situ measurements*

*The authors claim that calculating a correction factor for the chemiluminescent in-situ data by taking the average ratio of chemiluminescent to photolytic measurements. This would only work properly if the reasons for the high bias of the former instruments were identical at all measurement stations. While it is true that this correction does not influence the relative trends, the authors should at least mention this.*

**Thank you. We now clarify this in Section 2.1, as follows:**

"We correct the chemiluminescent NO₂ data by the observed ratio assuming that the inter-annual change is small and the high bias of the chemiluminescent measurements is identical at all sites."

*5 Importance of individual sources of AMF uncertainty*

*04/28: The authors claim that the first two factors are most important for the NO₂ trend analysis, but fail to back up their claim.*

**We revised this sentence as follows:**

"We find that the  NO₂ VCD trend analysis is particularly sensitive to the first two factors and we will discuss these in the following sections."

*6 Time span of lightning filter*

*06/20: The authors' choice of lightning filter (72hrs / 90km) seems arbitrary and needs to be justified. As the authors correctly state, the lifetime of NOx in the free troposphere can reach up to one week. By making their filter only 90km wide, a back-of-the envelope calculation quickly shows that the NOx produced by a single lightning occurrence can easily be transported considerably further within 72hrs than only 90km. The authors seem to be aware of this inconsistency, because they introduce an additional filter for the Northeast which depends on lightning occurrence in the South, implying a transport distance of many hundreds of kilometers.*

**We have stated this in Section 2.3.3, "**Since lightning usually occur along the track of a thunderstorm, the 90 km radius is more a constraint on lightning NOx effects across the track. The extended period of 72 hours

is to ensure that we exclude data affected by lightning NOx.**". We chose such filter constraints in order to balance between data availability and validity. The current constrains of 72 hours and a 90 km radius can ensure enough data (remove 2-27% data). Increasing the radius greatly will remove too much data. We discussed that our lightning filter is crude and needs improvements.**

*7 Minor comments*

*• 04/18: NO2 partial VCDs*

**Revised as suggested.**

*• 05/06: Which trends? Those with the default albedo, or those with the update? . . . )*

**Revised as follows:**

"The derived tropospheric NO2 VCD relative trends with default surface reflectance are referred as "Standard"."

*• 05/09: I personally find the name ocean trend misleading, as it has nothing to do with the ocean (except for the geographical location of the clean background region). Maybe the authors can come up with a name that somehow indicates the origin of the trend (e.g., instrument drift).*

**The relative trend in remote ocean potentially may stem from the similar reasons as the "hot" pixels of OMI. However, the nature of this trend is not fully understood. Thus, we decide to use the term 'ocean trend' to indicate that this trend is calculated at clean ocean region.**

*• 07/10: It seems that there are four different OMI-based NO2 trends*

**We revised this sentence as follows:**

"The ocean trend removal, MODIS albedo update, and lightning filter are then added in sequence to compute three different OMI-based $NO_2$ trends (in addition to "Standard") to compare to the AQS in situ results."

*• 07/15: To avoid confusion, please explicitly mention that these are absolute differences of the relative trends.*

**We now clarify this as follows:**

"OMI-based trends generally underestimate the decreasing trends by up to 3.7% yr$^{-1}$ (the absolute difference between relative trends) except for the large overestimation in the Midwest and the Northeast regions during DJF."

*• 09/05: trends of OMI data are less — than what?*

**We revised this sentence to prevent ambiguity.**

"Without the lightning filter, AQS decreasing trends are stronger  than the decreasing trends of OMI data  (Fig. 7)."

*• 09/07: OMI VCDs are not overestimated when not filtering for lightning NOx - the lightning NOx is part of the VCD. It leads to worse agreement between OMI and AQS trends, but then again, these are two fundamentally different measures anyways.*

**Yes, but for trend analysis, lightning signals need to be removed because they are sporadic and mask out the trends due to anthropogenic emission changes in an unpredictable manner.**

*• 09/11: What is a reduction of decreasing surface trends? Misleading phrase, since the trends are decreasing trends to begin with. Maybe it'd be better to say stronger decreasing trends or something similar.*

**We revised the term accordingly.**

"Therefore, the  weaker decreasing surface trends likely reflects a reduction of low-pressure dilution effect."

*• 09/12: Again, OMI VCDs are not biased due to lightning, see above.*

**We see the reviewer's point here. We clarified the discussion by adding** "While lightning NOx is part of OMI $NO_2$ observations, we treat the influence of lightning on the OMI tropospheric VCD trend as a bias for comparison purposes in this study since AQS data are not as strongly affected by lightning."

**We added in the conclusions** "While lightning NOx is part of OMI $NO_2$ observations, we treat the influence of lightning on the OMI tropospheric VCD trend as a bias for comparison purposes in this study since AQS data are not as strongly affected by lightning. Furthermore, lightning NOx effects need to be removed when using satellite observations to understand the effects of changing anthropogenic emissions."

*• 09/13: reduction of decreasing trends — see above*

**We revised the term accordingly.**

"Similarly, as anthropogenic emissions decrease, the positive bias of tropospheric VCDs due to lightning NOx becomes larger, likely resulting in  weaker decreasing trends."

*• 09/15: OMI VCDs are not wrong when they include lightning NOx – the authors should therefore not make the qualitative statement corrected here. Filtered would be better.*

**Revised as suggested.**

"We consider the lightning effects on surface NO2 trends to be mostly meteorological driven not by lightning NOx directly (e.g., Ott et al., 2010; Lu et al., 2017) and hence the  filtered OMI $NO_2$ data are likely

closer to emission related concentration changes."

*• 09/29: I would assume that the driving factor in stronger decreasing trends close to anthropogenic source regions is the decreasing emissions in those, resulting in less transported NOx in those areas.*

**If the atmospheric lifetime of NOx is the spatially homogeneous and the predominant source is anthropogenic, the reduction of NOx emissions will result in the same reduction (relative change) of NOx concentrations in rural and urban regions alike. The chemical non-linearity leads to changing atmospheric lifetime of NOx in response to NOx emission changes and subsequent more evident reduction in urban areas. Also, in rural areas where the dominant NOx sources are biogenic, such as, soil and lightning, the NOx concentration relative changes will be smaller if the reduction (relative change) of NOx emissions are the same. We have already stated these in Section 3.2:** "The larger decrease near the anthropogenic source regions reflect in part the nonlinear photochemistry (Gu et al., 2013) and in part to a stronger influence of NOx sources such as soils in rural regions."

*• 10/03: Since comparing VCDs to surface concentrations is a difficult issue to begin with, I would not blame the OMI retrievals for the differences – when comparing apples and oranges, why should one blame one and not the other for the differences? Saying that the OMI data are not designed for trend analysis doesn't make sense. If one has to design a dataset in order to be able to do trend analysis, maybe there just are no significant trends in the underlying data to begin with?*

**We have to disagree. All observation data need to be corrected when they are used for trend analysis. OMI retrievals are no different. It is more complex than, say, global surface temperature data because the retrieval is much more complex. Even for something as simple as surface temperature, one must be very careful when using the observation-based trends (especially for the early part of the dataset).**

*• In Fig. 1a-d, it is not clear if positive numbers mean that the OMI trend or the AQS trend is higher. Please update the Figure caption with a mathematically precise description (e.g., "OMI relative trend minus AQS relative trend").*

**We updated the caption as follows:**

"Panel (a) through (d) show the regional difference (OMI-based relative trends minus AQS relative trends) of annual relative trends between coincident OMI-based and AQS in situ data."

*• Fig. 3: Please update the Figure caption with a precise indication of the units, e.g., "number of days [. . . ] per REAM grid cell". Also, please spell out cloud-to-ground instead of CG in the caption.*

**Revised as suggested.**

"Number of days with NLDN detected cloud-to-ground (CG)  lightning per model grid cell per year during 2005-2014. The lightning occurrences are calculated using the REAM grid resolution."

*• Fig. 6: Please indicate NO2 somewhere in the Figure caption. Also, the legend for the OMI data should be something like OMI (lightning filter); after all, the data show trends of OMI NO2 columns and not of the lightning filter.*

**We revised the legend and the caption.**

"Seasonal relative trends of NO2 calculated from the AQS in situ measurements ("AQS", black line) and those derived from OMI data after applying the lightning filter ("OMI (lightning filter)", red line). The error bars represent 95th percentile confidence intervals. The coincident data points are less than those used in Figure 5 and therefore the AQS trends are not the same. "

*• Fig. 7: I don't understand what the figure legends are the same as in Figs. 6 and 8 is supposed to mean. Please clarify.*

*• Fig. 7: Please explicitly indicate in the Figure caption if statistically insignificant trends are shown or not.*

**We corrected the typos regarding the Figs. 4 and 6. We further clarified the confidence intervals of the error bars as follows:**

"The error bars represent 95th percentile confidence intervals. The relative trends are shown in Figs. 4 and 6. The figure legends are the same as in Figs. 4 and 6 but with the AQS trends subtracted from the OMI-based trends."

*• Fig. 9a: There is something wrong with the Figure caption, it does not contain a complete sentence (maybe there's just a of missing?). Please indicate what the barbs on the individual data points mean.*

**Revised as follows:**

"(a) The "Lightning filter" OMI-based NO$_2$ relative trend as a function 2005-2014 averaged OMI tropospheric NO2 VCD… The error bars represent 95th percentile confidence intervals. The red line shows a least-squares regression."

*• Fig. 9b: Please be specific about which OMI NO2 data you show in this Figure, using the nomenclature from earlier. As explained above, the notion of corrected is misleading.*

**Revised as follows:**

[revised manuscript text omitted]

---

## Referee Report (RR1)

Review of "Reconciling the differences between OMI-based and EPA AQS in situ NO2 trends"

After reviewing the responses by the authors and the revised manuscript, I do not recommend the manuscript for a publication to AMT in the present form. The responses to the reviews are not serious. There is no single analysis to support their responses.

The $NO_2$ concentrations measured at the surface monitors can be substantially different for the sites very close (e.g., 500 m). The size of OMI swath is 24 km x 13 km at the finest resolution and is often larger than this. In addition to differences in the spatial resolution, there are uncertainties in the satellite $NO_2$ retrievals and surface measurements. The trends of OMI tropospheric $NO_2$ columns and those of $NO_2$ measured at surface monitor can be similar as shown in the previous publications. But reconciling the differences between OMI and EPA AQS $NO_2$ trends for large regions can not be a measure for improvement of OMI NO2 retrievals and their trends.

Overall, the manuscript needs a major revision if the authors would like to publish it at AMT. I list my suggestions below.

(1) Authors need to make their focus clear. OMI data co-located with the AQS are mainly discussed through the manuscript, but all of sudden the trends of OMI $NO_2$ for large regions are emphasized (e.g., abstract line 27-31). The comparison of the trends of OMI $NO_2$ for large regions with the trends from the AQS does not make sense. It is confusing if the authors mention the OMI trends at the AQS or the OMI trends for large regions such as West, Midwest etc for all the figures and the text.

(2) Please make careful statements based on clear or enough proofs. The authors added "However, the current OMI tropospheric NO2 retrievals are not designed for analyzing multi-year tropospheric NO2 trends". I do not understand what it actually means. Does it mean the previous publications on OMI $NO_2$ trends are not correct?

(3) The ocean trends do not look significant as another colleague mentioned. Albedo correction based on MODIS data looks promising as Russell et al. (2011) demonstrated. Lightning filtering also gives new insights for southern US. It is important to show spatial variability of the trends or $NO_2$ columns from adopting MODIS albedo and lightning filtering similar to Figure 8. Detailed spatial distribution rather than the simple values for 4 large districts would be useful. Add explanations of why the impact of lightning filtering is large for the Northeast US (not only the South US, see Figure 1).

(4) Referring to Lamsal et al. (2015), the authors only mentioned average values. Lamsal et al. (2015) also stated that the impact of changing anthropogenic emissions in calculating a priori profiles can be large up to 15% more reductions in the declining trend depending on the location. Lamsal et al. used 1 degree x 1.25 degree GMI model grid resolution to produce trace gas profiles. The authors have the REAM model setting with the 36 km resolution and are capable of producing own profiles rather than fully depending on discussions in the previous publication. I am not convinced with lines 28-29, page 4, "The $NO_2$ VCD trend analysis is particularly sensitive to the first two factors and we will discuss these in the following sections".

(5) Discrepancies between $NO_2$ chemiluminescence to photolytic converter measurement are small in the morning (7-9 am) and become larger in the afternoon. The plots of diurnal variations of the ratios at each station would give a confidence in the quality of the measurements.

(6) I do not understand the meaning of Figure 9. Is this for the apple-orange comparison of the trends from the AQS and those from OMI data for the large areas?

---

## Editor Decision (ED1)

Dear Ruixiong,

I have now received reviews of your revised manuscript from both reviewers. One of the reviewers has entered his comments in the wrong field so I attach it below for your reference.

As you will see, both reviewers are not at all satisfied with the manuscript and the changes you made. After reading the manuscript again carefully, I agree with their assessment: the manuscript as it is cannot be published in AMT.

There are three main reasons for this conclusion:

1) The manuscript lacks important information. Even after reading it several times, it is for example not clear to me

- If and how the surface measurements were sampled in time to match the overpass of the OMI data
- If and how the OMI data were sampled in space to match the surface data
- Whether a constant row filter removing rows 26-55 was used or if the (time dependent) flagging in the DOMINO product was used (both is stated in the text).
- How exactly the drift over the ocean was accounted for in the data analysis
- Which version of the AMF lookup table was used (you make reference to a recent paper by Lorente et al. for DAK but it is not clear if you use the DOMINO LUT or a more recent version from the QA4ECV project)

2) The explanations given in the manuscript for some of the observed effects are unclear or not convincing:

- The offset correction using data over the ocean is an interesting idea and is in fact used in many satellite data products. However, here the tropospheric slant column is used, which is the result of subtracting the assimilated stratospheric $NO_2$ column from the retrieved slant column. Any global drift in the OMI $NO_2$ slant column data would be absorbed by the data assimilation. The remaining (very small) drift is probably indicative of some problems in the stratospheric correction which would also explain the large variability (note that even negative columns occur in some months) and is certainly not representative for offsets in $NO_2$ slant columns over all of the US. This is the reason why the authors do not apply monthly corrections which they should if they would trust the offset correction approach.
- The explanation given for the offset trend (increased striping) does not make sense for two reasons: First, striping is by definition deviation from the mean value and thus should not contribute to a drift in the mean, and second, I assume that the authors use destriped data (but also this information is not given in the manuscript).
- The albedo trend correction is an interesting result, but little is said about what the reasons for the differences between MODIS and TEMIS albedo values are. It also is not clear to me if MODIS albedo trends of the order of 0.5% per year are significant.
- The lightning filter makes a lot of sense if satellite and surface observations are to be compared. However, the results are puzzling and in contrast to the conclusions drawn in the manuscript: As is evident from Table 1, surface station trends are more affected by the lightning filter than OMI data with the exception of the West, where only little lighting is found over the regions having $NO_2$ above the threshold of 1E15 molec/cm2. The clear

improvement in consistency of the two data sets in the South when applying the filter is mainly the result of a change in in-situ trend! In that sense, the lightning filter is more a filter on the surface data than on the OMI data. This result points at sampling issues in the surface data, not inadequacy of the standard OMI product.

3) The underlying assumption of the whole manuscript is, that trends of surface NO2 in-situ observations should agree with OMI NO2 column trends. As both reviewers point out, this appears to be an apple and oranges comparison, and while it is interesting to compare these two trends, it is by no means clear that they should be exactly the same. As in part pointed out in the manuscript, there are several reasons why they could be different, including representativity of the measurement location (see Fig. 9), vertical NO2 distribution, non-linearity of NOx chemistry and different sensitivity to changes in boundary layer height.

In my personal opinion, the better agreement between surface in-situ trends and the OMI trends after applying the corrections you suggest is a) a coincidence and b) not significant within the uncertainties of the method and the comparison.

In order to make this manuscript publishable in AMT, you need to

- Add the missing information and clarify the details of your approach
- Reconsider your discussions, explanations and conclusions in view of the comments made by the reviewers and listed above (or convince me that I have not understood your arguments)
- Rephrase the manuscript in order not to oversell the achieved improvement in consistency as proving that your corrected OMI time series is more correct than the original one, unless you can show that this is the case.

These are major revisions and as both reviewers have declined to see another version of this manuscript, I will have to be satisfied with the next version or I will have to reject the paper.

Best regards,

Andreas

===================== Report of anonymous referee # 1 ================================

Unfortunately, the authors didn't change the manuscript in several passages that could have been clearer. Also, I very strongly disagree with the authors' comment that "All observation data need to be corrected when they are used for trend analysis." In its generalness, this is not scientific, in my opinion. This is only the case if one wants to extract information from measurements that was not originally measured, such as comparing apples and oranges. Also, the manuscript still leaves the impression that it is "wrong" of the OMI VCD trends to not be identical to the in-situ measurement network trends, which is still a comparison of apples and oranges in my view.

Having said that, I don't see any willingness by the authors to accept my views or change the manuscript accordingly (which would be possible without harm to the original message / scientific content of the manuscript). Unfortunately, the authors chose not to make it more balanced paper. I still don't like the paper very much, and find it to be rather unbalanced.

However, given that I only asked for "minor revisions" in the first place, and that the authors did address the most severe points, I don't see any grounds to not publish the article. I wouldn't know what to ask for in a potential revision of the manuscript -- I could raise the same points again as I raised in the original review, and the authors still wouldn't agree.

 Given that the actual technical modifications to the OMI retrieval are interesting and should be made public to the scientific community, I don't oppose publishing the paper. I just don't like it.

=============================================================================

---

## Author Response (AR2)

We thank the editor and the two reviewers for their thoughtful comments and suggestions concerning our research. We made major revisions accordingly. We present our response and changes below. The editor's and the reviewers' comments and suggestions are shown in *italic*. Authors' responses are in **bold**.

**Response to Editor:**

*There are three main reasons for this conclusion:*

*1) The manuscript lacks important information. Even after reading it several times, it is for example not clear to me*

*• If and how the surface measurements were sampled in time to match the overpass of the OMI data*

*• If and how the OMI data were sampled in space to match the surface data*

**We only used the OMI pixels which cover the corresponding AQS sites. The in situ NO$_2$ data are temporally interpolated based on the overpassing time of the corresponding OMI pixels. We now clarify this in Section 3:**

"We group the analysis results into different regions: (a) West, (b) Midwest, (c) Northeast, and (d) South (Fig. 1), following the regional divisions by the United States Census Bureau. To make a fair comparison between the in situ and OMI-based trends, we only use spatially and temporally coincident in situ and OMI NO$_2$ observations in Section 3.1. The AQS data are temporally interpolated based on the overpassing time of the available OMI pixels which cover the corresponding AQS sites. Similarly, only OMI pixels covering the corresponding available AQS sites are used. The data from each dataset are then aggregated and averaged on a regional basis into time series to calculate the regional trends."

*• Whether a constant row filter removing rows 26-55 was used or if the (time dependent) flagging in the DOMINO product was used (both is stated in the text).*

**We use a constant row filter removing rows 1-5 and rows 26-60 in the analysis. We now clarify this in Section 2.3.1:**

"For trend and other analyses of OMI tropospheric VCDs, the data of anomalous pixels must be removed. The row anomaly initially occurred in June 2007 and subsequently in later years affected rows 26-40 (Schenkeveld et al., 2017). Additional anomalies can be found in some years in rows 41-55. For trend analysis from 2005-2014, we exclude rows 26-55, consistent with our understanding of the row anomaly (Schenkeveld et al., 2017) . In addition, the data of coarse spatial resolution from rows 1-5 and rows 56-60 are also excluded, as suggested by Lamsal et al. (2015). The selection of rows 6-25 used in this research is stricter than the data flags in the DOMINO v2 product. Furthermore, we exclude OMI data with cloud fraction > 0.3 to minimize retrieval uncertainties due to clouds and aerosols (Boersma et al., 2011; Lin et al., 2014)."

*• How exactly the drift over the ocean was accounted for in the data analysis*

**We calculate the influence of ocean trend (drift) for each year and subtract it from OMI tropospheric SCDs in the data analysis. We now clarify this in Section 2.3.1:**

"However, removing this background ocean (absolute) trend has a non-negligible effect in reducing the OMI relative trend (Fig. 1). We treat this trend as a systematic bias. We calculate the contribution from the ocean (absolute) annual trend to SCDs for each year and we subtract it from OMI tropospheric NO$_2$ SCDs uniformly in the following analysis. Since the origin of this trend is not yet clear, the ocean trend removal method may need updates in future studies."

*• Which version of the AMF lookup table was used (you make reference to a recent paper by Lorente et al. for DAK but it is not clear if you use the DOMINO LUT or a more recent version from the QA4ECV project)*

**We use the DOMINO LUT and we update the reference in Section 2.3:**

"As the vertical distribution of $NO_2$ is usually unknown, we typically substitute $x_l$ by an a priori profile ($x_{l,apriori}$) from a CTM. $AMF_l$ is the sensitivity of $NO_2$ SCD to VCD at a given altitude (Eskes and Boersma, 2003), and is computed using the Double Adding KNMI (DAK) RTM ( )."

*2) The explanations given in the manuscript for some of the observed effects are unclear or not convincing:*

*• The offset correction using data over the ocean is an interesting idea and is in fact used in many satellite data products. However, here the tropospheric slant column is used, which is the result of subtracting the assimilated stratospheric NO2 column from the retrieved slant column. Any global drift in the OMI NO2 slant column data would be absorbed by the data assimilation. The remaining (very small) drift is probably indicative of some problems in the stratospheric correction which would also explain the large variability (note that even negative columns occur in some months) and is certainly not representative for offsets in NO2 slant columns over all of the US. This is the reason why the authors do not apply monthly corrections which they should if they would trust the offset correction approach.*

**As stated previously, we now state in the paper that "**…the origin of this trend is not yet clear…**". The exact origin of the ocean trend is unknown and needs further investigation. We consider the ocean trend to be a systematic bias and we remove this ocean trend in our analysis. We now emphasize this in Section 2.3.1:**

"…This trend may  the inaccurately simulated stratospheric SCDs (A. Richter, personal communication, 2018) or the increase in the magnitude of the stripes (step-wise SCD variability from one row to another) in time, which originates from the use of a constant (2005-averaged) solar irradiance reference spectrum in the DOAS spectral fits throughout the mission and the weak increase of noise in the OMI radiance measurements (K. F. Boersma, personal communication, 2017; Zara et al., 2018)… However, removing this background ocean (absolute) trend has a non-negligible effect in reducing the OMI relative trend (Fig. 1). We treat this trend as a systematic bias. We calculate the contribution from the ocean (absolute) annual trend to SCDs for each year and we subtract it from OMI tropospheric $NO_2$ SCDs uniformly in the following analysis. Since the origin of this trend is not yet clear, the ocean trend removal method may need updates in future studies."

**As stated in Section 2.3.1, the results using monthly corrections yield similar results and same conclusions. We add Figures S2 and S3 in the supplement, i.e. the monthly ocean trends and Figure 7 replotted using data with monthly corrections. The monthly ocean trends are calculated using the Mann-Kendall method and are large during SON and DJF (2.0 to 5.2 x10$^{13}$ molecules cm$^{-2}$ yr$^{-1}$) and small during MAM and JJA (-0.6 to 1.1 x10$^{13}$ molecules cm$^{-2}$ yr$^{-1}$). The use of the monthly corrections and yearly corrections lead to the same conclusions, as illustrated partly by comparing Figure S3 and Figure 7. We use the (absolute) annual VCD corrections in the manuscript.**

**We feel the editor has a good point on the unknown nature of the ocean VCD trend, we added in the conclusions section: "**Among the corrections, the background ocean trend removal is not as significant as the latter two. Since the origin of this trend is not yet clear, the ocean trend removal method may need updates in future studies.**"**

*• The explanation given for the offset trend (increased striping) does not make sense for two reasons: First, striping is by definition deviation from the mean value and thus should not contribute to a drift in the mean, and second, I assume that the authors use destriped data (but also this information is not given in the manuscript).*

**Our coauthor K. F. Boersma suggested that striping could be the reason for the background ocean trend. He indicated that the increasing deviation from the mean (striping) might lead to a biased ocean trend from using a constant solar radiance spectrum in the DOAS spectral fitting (K. F. Boersma, personal communication, 2017). The section is changed to:**

"This trend may reflect the inaccurately simulated stratospheric SCDs (A. Richter, personal communication, 2018) or the increase in the magnitude of the stripes (step-wise SCD variability from one row to another) in time, which originates from the use of a constant (2005-averaged) solar irradiance reference spectrum in the DOAS spectral fits throughout the mission and the weak increase of noise in the OMI radiance measurements (K. F. Boersma, personal communication, 2017; Zara et al., 2018)."

**As indicated in the response above, we do not know the exact nature of this trend. We use the OMI data without destriping through the analysis. We now clarify this in Section 2.3:**

"We retrieve the tropospheric $NO_2$ VCDs using the tropospheric slant column densities (SCDs, without destriping) from the Royal Dutch Meteorological Institute (KNMI) Dutch OMI $NO_2$ product (DOMINO v2, Boersma et al., 2011)."

*• The albedo trend correction is an interesting result, but little is said about what the reasons for the differences between MODIS and TEMIS albedo values are. It also is not clear to me if MODIS albedo trends of the order of 0.5% per year are significant.*

**We focus on the relative trends instead of the absolute values of the albedo values. The albedo data used in original OMI DOMINO v2 product incorporates a climatology dataset with snow/ice albedo adjustment (Section 2.3.2). As a result, DOMINO v2 albedo exhibits no trends in places without snow/ice such as the West and the South (Figure 5). The snow/ice albedo adjustment raises the albedo to 0.6 for pixels whenever and wherever snow is detected by NISE dataset. Therefore, the discrepancies between two albedo datasets mainly come from: (1) the temporal variations of albedo (climatology data from DOMINO v2 vs 16-day data by MODIS); (2) the albedo in case of snow or ice (snow/ice adjustment by DOMINO v2 vs the observations by MODIS). We now clarify this in Section 3.1.2, as follows:**

"The OMI DOMINO v2 incorporates a climatology albedo dataset (Kleipool et al., 2008) with snow/ice albedo adjustment , in which the albedo value is reset to be 0.6 if snow/ice is reported in the NASA Near-real-time Ice and Snow Extent (NISE) dataset (Boersma et al., 2011). The climatology albedo data  have no trends. Thus, the trends of albedo from the DOMINO v2 product mainly originate from the yearly variation of NISE detected snow/ice and to a lesser extent the OMI sampling variation. The noticeable seasonal trend of the OMI DOMINO v2 albedo dataset is the 3.9% $yr^{-1}$ increase in DJF of the Midwest and a smaller DJF increase (1.0%) of the Northeast. In contrast, the MODIS albedo dataset exhibits a smaller positive DJF trend (0.8% $yr^{-1}$), 3.1% $yr^{-1}$ less than the trend from DOMINO v2, in the Midwest, and a small negative DJF trend (-0.8%) in the Northeast. These differences suggest that using a fixed snow/ice albedo and climatology albedo data are inadequate. The comparison to the AQS data shows that the MODIS albedo update leads to better agreement between satellite and in suit trends in winter in these regions (Fig. 4)."

**The differences between relative trends of the two albedo datasets in the West and the South are about 0.5-1% $yr^{-1}$. These differences could result from the temporal variation of surface albedo and do not contribute significantly to the differences between relative trends of OMI and AQS $NO_2$, as shown in Figure 4. The differences between two albedo datasets in the Midwest and the Northeast mainly come from the inaccurate snow/ice albedo adjustments and are most significant during DJF (Figure 4).**

*• The lightning filter makes a lot of sense if satellite and surface observations are to be compared. However, the results are puzzling and in contrast to the conclusions drawn in the manuscript: As is evident from Table 1, surface station trends are more affected by the lightning filter than OMI data with the exception of the West, where only little lighting is found over the regions having NO2 above the threshold of 1E15 molec/cm2. The clear improvement in consistency of the two data sets in the South when applying the filter is mainly the result of a change in in-situ trend! In that sense, the lightning filter is more a filter on the surface data than on the OMI data. This result points at sampling issues in the surface data, not inadequacy of the standard OMI product.*

**We agree that the lightning filter is most useful when comparing trends from the two datasets. We now state it clearly in the abstract:** "We recommend future studies to apply these procedures (ocean trend removal and MODIS albedo update) to ensure the quality of satellite-based NO2 trend analysis,  and apply the lightning filter in  studying surface NOx emission changes using satellite observations and in comparison with the trends derived from in situ NO$_2$ measurements."

**The lightning filter affects the sampling of AQS data as well as the OMI data. A similar effect can be seen when comparing AQS data with corresponding OMI data used in the study and all AQS data. It is not surprising since while lightning NOx does not affect surface AQS sites directly, chemical (such as cloud effects on photolysis rates, lifetime of NOx, and the ratio of NO$_2$/NO) and physical (transport by advection and convection) processes affect surface NO$_2$ concentrations too. Added in Section 3.2:**

"Table 1 shows the effects of data sampling when both AQS and OMI data are analyzed and when the lightning filter is applied."

*3) The underlying assumption of the whole manuscript is, that trends of surface NO2 in-situ observations should agree with OMI NO2 column trends. As both reviewers point out, this appears to be an apple and oranges comparison, and while it is interesting to compare these two trends, it is by no means clear that they should be exactly the same. As in part pointed out in the manuscript, there are several reasons why they could be different, including representativity of the measurement location (see Fig. 9), vertical NO2 distribution, non-linearity of NOx chemistry and different sensitivity to changes in boundary layer height.*

*In my personal opinion, the better agreement between surface in-situ trends and the OMI trends after applying the corrections you suggest is a) a coincidence and b) not significant within the uncertainties of the method and the comparison.*

**We agree that the relative trends from NO$_2$ VCDs and surface concentrations should not be exactly the same considering the different spatial sampling, inherent difference between trends from NO$_2$ tropospheric VCDs and surface concentrations, chemical non-linearity, effects of emission changes on NO$_2$ profiles and etc. The focus of this research is to quantify what the difference between relative trends from both datasets are and how the ending results help us understand the surface emission changes. To avoid confusion, we change our title to "**Comparing OMI-based and EPA AQS in situ NO$_2$ trends: Towards understanding surface NOx emission changes**". We modify our manuscript accordingly.**

*In order to make this manuscript publishable in AMT, you need to*

*• Add the missing information and clarify the details of your approach*

*• Reconsider your discussions, explanations and conclusions in view of the comments made by the reviewers and listed above (or convince me that I have not understood your arguments)*

*• Rephrase the manuscript in order not to oversell the achieved improvement in consistency as proving that your corrected OMI time series is more correct than the original one, unless you can show that this is the case.*

**We have modified the manuscript as suggested.**

**Response to Referee #1:**

*Also, I very strongly disagree with the authors' comment that "All observation data need to be corrected when they are used for trend analysis." In its generalness, this is not scientific, in my opinion. This is only the case if one wants to extract information from measurements that was not originally measured, such as comparing apples and oranges. Also, the manuscript still leaves the impression that it is "wrong" of the OMI VCD trends to not be identical to the in-situ measurement network trends, which is still a comparison of apples and oranges in my view.*

**We made substantial changes to the manuscript as described in the responses to the Editor's questions and believe that we addressed the reviewer's main concern.**

**Response to Referee #2:**

*The NO2 concentrations measured at the surface monitors can be substantially different for the sites very close (e.g., 500 m). The size of OMI swath is 24 km x 13 km at the finest resolution and is often larger than this. In addition to differences in the spatial resolution, there are uncertainties in the satellite NO2 retrievals and surface measurements. The trends of OMI tropospheric NO2 columns and those of NO2 measured at surface monitor can be similar as shown in the previous publications. But reconciling the differences between OMI and EPA AQS NO2 trends for large regions can not be a measure for improvement of OMI NO2 retrievals and their trends.*

**We made substantial changes to the manuscript as described in the responses to the Editor's questions and believe that we addressed the reviewer's concern.**

*Overall, the manuscript needs a major revision if the authors would like to publish it at AMT. I list my suggestions below.*

*(1) Authors need to make their focus clear. OMI data co-located with the AQS are mainly discussed through the manuscript, but all of sudden the trends of OMI NO2 for large regions are emphasized (e.g., abstract line 27-31). The comparison of the trends of OMI NO2 for large regions with the trends from the AQS does not make sense. It is confusing if the authors mention the OMI trends at the AQS or the OMI trends for large regions such as West, Midwest etc for all the figures and the text.*

**Table 1 shows the effects of sampling for regional trends. For comparison purposes, we need to use co-located data; but for understanding regional trends, we need to use all data. For readers interested in satellite data applications, they would be interested in co-located data. But general readers they would be interested in the trends of all data.**

*(2) Please make careful statements based on clear or enough proofs. The authors added "However, the current OMI tropospheric NO2 retrievals are not designed for analyzing multi-year tropospheric NO2 trends". I do not understand what it actually means. Does it mean the previous publications on OMI NO2 trends are not correct?*

**We removed this sentence.**

*(3) The ocean trends do not look significant as another colleague mentioned. Albedo correction based on MODIS data looks promising as Russell et al. (2011) demonstrated. Lightning filtering also gives new insights for southern US. It is important to show spatial variability of the trends or NO2 columns from adopting MODIS albedo and lightning filtering similar to Figure 8. Detailed spatial distribution rather than the simple values for 4 large districts would be useful. Add explanations of why the impact of lightning filtering is large for the Northeast US (not only the South US, see Figure 1).*

**We now state in the manuscript that the effects of removing ocean trends are not as large as albedo and lightning filter processing. We present Figure R1 here with the spatial distributions of OMI-based $NO_2$ trends of the standard product and the three variants.**

[Figure]

**Figure R1: Annual relative trends of OMI-based NO₂ for "Standard" (a), "Ocean" (b), "MODIS" (c), and for "Lightning filter" (d) as the colored background. Black bordered circles indicate corresponding AQS NO₂ trends. Grid cells with 2005-2014 mean NO₂ VCDs < 1x10^15 molecules cm^-2 are excluded in the analysis and are shown in white.**

**The Northeast U.S. can be affected by lightning NOx transported from the South. We design the lightning filter such that we screen the data affected by transported lightning NOx from the South to the Northeast. We have stated in Section 2.3.3,** "While there are fewer lightning flashes in the Northeast than the South (Fig. 3), large amounts of lightning NOx can be produced by high flash ratios of severe thunderstorms and they can be transported northward from the South to the Northeast (Choi et al., 2005). We therefore further filter OMI NO₂ data in the Northeast on the basis of CG lightning flash rates in the South. If the average CG flash rate in the South exceeds the 95th percentile value of the NLDN observations, which is 0.035 flash km^-2 day^-1 (Fig. S2 in the Supplement), we exclude in the analysis the Northeast OMI data in the following 72 hours." **We add the following clarification in Section 3.1.3:**

"Fig. 6 shows that the lightning filter significantly reduces the difference between the OMI-based relative trend and that of the AQS data by 0.5-1.4% yr^-1 in the Northeast and 0.9-1.3% yr^-1 in the South. As a result, the seasonal trend differences are within 0.9% yr^-1 in these two regions except during SON. The Northeast is affected by the lightning filter due to lightning in this region and transport of lightning NOx from the South (Section 2.3.3)."

*(4) Referring to Lamsal et al. (2015), the authors only mentioned average values. Lamsal et al. (2015) also stated that the impact of changing anthropogenic emissions in calculating a priori profiles can be large up to 15% more reductions in the declining trend depending on the location. Lamsal et al. used 1 degree x 1.25 degree GMI model grid resolution to produce trace gas profiles. The authors have the REAM model setting with the 36 km resolution and are capable of producing own profiles rather than fully*

*depending on discussions in the previous publication. I am not convinced with lines 28- 29, page 4, "The NO2 VCD trend analysis is particularly sensitive to the first two factors and we will discuss these in the following sections".*

**Lamsal et al. (2015) illustrated in the Figure 3 of the referenced publication that the difference in reduction is generally less than 2% in $NO_2$ abundant regions during 2005-2010 (equivalent to 0.3% $yr^{-1}$). On a regional basis, it is difficult that the a priori profile can have a significant effect on the relative trends. As reported in a newly published study, the reduction of emission may have slowed down after 2010 (Jiang et al., 2018). This would lead to an even smaller contribution from NOx emission changes to $NO_2$ trend during the studied period of 2005-2014. To fully quantify the emission changes, inverse modeling of NOx emission over a 10-year timespan is needed.**

**We update the manscript as below:**

" In this study, we find that the first two factors are essential in $NO_2$ VCD trend analysis and we will discuss these in the following sections."

*(5) Discrepancies between NO2 chemiluminescence to photolytic converter measurement are small in the morning (7-9 am) and become larger in the afternoon. The plots of diurnal variations of the ratios at each station would give a confidence in the quality of the measurements.*

**We have stated in Section 2.1,** "In this study, we only examine the relative trends and therefore the analysis results are not affected by the uncertainties in the in situ $NO_2$ measurement corrections.". **Due to data availability, only one site has more than one year of consecutive data with both measurements. We show the averaged diurnal cycle of ratios in Figure R2, as below.**

[Figure]

**Figure R2. The diurnal cycle of ratios between $NO_2$ surface concentrations of chemiluminescent to photolytic instruments in Sacramento, CA. The errorbars represent standard deviations.**

*(6) I do not understand the meaning of Figure 9. Is this for the apple-orange comparison of the trends from the AQS and those from OMI data for the large areas?*

**Figure 9a shows the "Lightning filter" OMI-based NO$_2$ relative trends as a function of averaged NO$_2$ VCDs. Figure 9b shows the averaged NO$_2$ VCDs across U.S. Figure 9 intends to help readers understand that the NO$_2$ reduction is larger in NO$_2$ abundant areas where most AQS sites are located. For general readers, they will want to understand how the regional trends of OMI data compare to AQS data. We show that the difference is very large and explain the reasons.**

**REFERENCES**

Jiang, Z., McDonald, B. C., Worden, H., Worden, J. R., Miyazaki, K., Qu, Z., Henze, D. K., Jones, D. B. A., Arellano, A. F., Fischer, E. V., Zhu, L., and Boersma, K. F.: Unexpected slowdown of US pollutant emission reduction in the past decade, Proceedings of the National Academy of Sciences, 10.1073/pnas.1801191115, 2018.

[revised manuscript text omitted]